# Extreme temperature and precipitation response to solar dimming and stratospheric aerosol geoengineering

Duoying Ji[1], Songsong Fang[1], Charles L. Curry[2,3], Hiroki Kashimura[4], Shingo Watanabe[5], Jason N. S. Cole[6], Andrew Lenton[7], Helene Muri[8,9], Ben Kravitz[10] and John C. Moore[1,11,12]

[1]College of Global Change and Earth System Science, Beijing Normal University, Beijing 100875, China
[2]School of Earth and Ocean Sciences, University of Victoria, Victoria, British Columbia, Canada
[3]Pacific Climate Impacts Consortium, University of Victoria, Victoria, British Columbia, Canada
[4]Department of Planetology/Center for Planetary Science, Kobe University, Kobe, Japan
[5]Japan Agency for Marine-Earth Science and Technology, Yokohama, Japan.
[6]Canadian Centre for Climate Modelling and Analysis, Environment and Climate Change Canada, Victoria, British Columbia, Canada
[7]Oceans and Atmosphere, Hobart, Tasmania, Australia
[8]Department of Geosciences, University of Oslo, Oslo, Norway
[9]Department of Energy and Process Engineering, Norwegian University of Science and Technology, Trondheim, Norway
[10]Atmospheric Sciences and Global Change Division, Pacific Northwest National Laboratory, Washington, USA
[11]Arctic Centre, University of Lapland, P.O. Box 122, 96101 Rovaniemi, Finland
[12]CAS Center for Excellence in Tibetan Plateau Earth Sciences, Beijing 100101, China

*Correspondence to*: John C. Moore (john.moore.bnu@gmail.com)

**Abstract.**

We examine extreme temperature and precipitation under two potential geoengineering methods forming part of the Geoengineering Model Intercomparison Project (GeoMIP). The solar dimming experiment G1 is designed to completely offset the global mean radiative forcing due to a $CO_2$-quadrupling experiment (abrupt4×CO2), while in GeoMIP experiment G4, the radiative forcing due to the representative concentration pathway 4.5 (RCP4.5) scenario is partly offset by a simulated layer of aerosols in the stratosphere. Both G1 and G4 geoengineering simulations lead to lower minimum temperatures (TNn) at higher latitudes, and on land primarily through feedback effects involving high latitude processes such as snow cover, sea ice and soil moisture. There is larger cooling of TNn and maximum temperatures (TXx) over land compared with oceans, and the land-sea cooling contrast is larger for TXx than TNn. Maximum 5-day precipitation (Rx5day) increases over subtropical oceans,

whereas warm spells (WSDI) decrease markedly in the tropics, and the number of consecutive dry days (CDD) decreases in most deserts. The precipitation during the tropical cyclone (hurricane) seasons becomes less intense, whilst the remainder of the year becomes wetter. Stratospheric aerosol injection is more effective than solar dimming in moderating extreme precipitation (and flooding). Despite the magnitude of the radiative forcing applied in G1 being ~7.7 times larger than in G4, and differences in the aerosol chemistry and transport schemes amongst the models, the two types of geoengineering show similar spatial patterns in normalized differences of extreme temperatures changes. Large differences mainly occur at northern high latitudes, where stratospheric aerosol injection more effectively reduces TNn and TXx. While the pattern of normalized differences of extreme precipitation is more complex than that of extreme temperatures, generally stratospheric aerosol injection is more effective in reducing tropical Rx5day, while solar dimming is more effective over extra-tropical regions.

**1 Introduction**

Global atmospheric greenhouse gas (GHG) concentrations continue to increase due to slow progress in reducing net GHG emissions in the industrialized world. Even if countries with existing commitments reduce emissions to meet their national goals (or aspirational targets under the 2015 Paris Agreement), this may not be sufficient to avoid dangerous or irreversible climate change (Sanderson et al., 2016). Climate engineering is increasingly being discussed as a means to lessen or ameliorate the effects of global warming. In particular, Solar Radiation Management (SRM), the artificial reduction of incoming solar radiation has been increasingly studied: examples include mirrors in space (Mautner, 1989), stratospheric aerosol injection (e.g., Budyko, 1977; Crutzen, 2006) or marine cloud brightening (e.g., Latham, 1990). Scientific investigation of SRM has made use of several different climate models examining various degrees of SRM and greenhouse gas forcing (e.g., Bala et al., 2008; Irvine et al., 2011; Schmidt et al., 2012). While gross features of (for example) global temperature patterns under SRM appear robust, more subtle climate indices require standardized experimental design. Kravitz et al. (2011, 2013c) defined a set of numerical SRM experiments under the Geoengineering Model Intercomparison Project (GeoMIP), comprising solar dimming experiments (G1 and G2), stratospheric aerosol injection simulations (G3 and G4) and marine cloud brightening experiments (G4cdnc, G4sea-salt).

The mean climate response under G1 and G2 of diverse climate variables, e.g. temperature, precipitation, sea level pressure has been well described (e.g., Schmidt et al., 2012; Kravitz et al., 2013b; Tilmes et al., 2013; Jones et al., 2013). Curry et al. (2014) drew attention to the changes in temperature and precipitation extremes in models running the reduced solar radiation G1 experiment, and Aswathy et al.

(2015) examined extremes under G3 and G3-SSCE (marine cloud brightening by sea salt emission, modelled after GeoMIP experiment G3). Dagon and Schrag (2017) showed that solar geoengineering mitigates extreme heat events from greenhouse warming, though the regional response is variable in part due to varying soil moisture content: soils dry out over the course of the summer as daily maximum temperature increases, and this relationship is strengthened under solar geoengineering. These are the

only dedicated analyses of climate model extreme indices under geoengineering to date.

This paper will provide a first look at the difference in the extremes of temperature and precipitation between two geoengineering methods: G1 (solar dimming) and G4 (stratospheric aerosol injection) experiments. Both methods would cool Earth's surface by reducing sunlight reaching the surface, either by aerosols reflecting sunlight or by artificially reducing the solar constant in climate models. The

injected stratospheric aerosols under G4 not only scatter shortwave radiation, also absorb near infrared and longer wavelengths radiation (Lohmann and Feichter, 2005). The differences between stratospheric aerosol injection and solar dimming are influenced strongly by the absorption of longwave radiation by aerosols; this atmospheric heating imbalance could further stabilize the troposphere and lead to stronger precipitation reduction under stratospheric aerosol injection than under solar dimming (Niemeier et al.,

2013). That there can be a difference in the mean climate response in reduced solar constant and increased stratospheric sulphate aerosols has been shown previously (Yu et al., 2015; Niemeier et al., 2013; Ferraro et al., 2014) and we expect that this will also be evident in the temperature and precipitation extremes. We perform analyses on daily output from GeoMIP models that have completed both G1 and G4, which is a limited subset of models with several excluded from these analyses because only monthly resolution

output was saved. We take the results from G1 relative to its corresponding CMIP5 experiment, abrupt4×CO2 to examine impacts of solar dimming, and take the results from G4 relative to rcp45 (the simulations forced by the RCP4.5 scenario) as the impact of stratospheric aerosol injection. The paper is organized as follows: The multimodel ensembles and the definitions of indices are briefly described in Section 2. The probability density functions of monthly mean temperature and precipitation and the

results are given in Section 3, along with global mean time series and spatial and seasonal differences of

the extreme climate indices in the two SRM experiments. Finally, and a summary of the main findings and conclusion are given in Section 4.

## 2 Data and Methods

### 2.1 GeoMIP experiments

G1 simulates balancing the GHG forcing from the CMIP5 experiment abrupt4×CO2 (instantly quadrupled $CO_2$ relative to pre-industrial levels) by decreasing solar irradiance. The G1 experiment runs for 50 years beginning from the control run (the piControl scenario; Taylor et al., 2012). The globally averaged top of atmosphere (TOA) radiation differences between G1 and piControl are no more than 0.1 $Wm^{-2}$ (Kravitz et al., 2011). The G1 results can also be naturally compared with results from the

abrupt4×CO2 simulation itself, which most model groups have performed (Taylor et al., 2012).

G4 is based on the RCP4.5 future climate scenario (hereafter "rcp45", Meinshausen et al., 2011; Taylor et al., 2012), with additional injection of $SO_2$ into the tropical lower stratosphere at a rate of 5 Tg per year from the year 2020. The G4 experiments do not specify any specific treatment of chemical or physical properties, so inter-model differences are expected to be larger than in G1 simply from

differences in the implementation of the stratospheric aerosol injection (Kravitz et al., 2011; Yu et al., 2015). The stratospheric aerosol injection experiment, G4, is a much smaller signal with respect to its reference under the mild GHG forcing specified by RCP4.5 than the G1 experiment with respect to its reference abrupt4×CO2. The global temporally averaged forcing of the G1 solar dimming experiment ranges from -9.6 to -6.4 $Wm^{-2}$, and the G4 stratospheric aerosol injection experiment ranges from $-3.6$

to $-1.6$ $Wm^{-2}$, depending on the model (Schmidt et al., 2012; Kashimura et al., 2017).

We analyze the daily output from six Earth system models, which completed both the G1 and G4 experiments (Table 1). In order to compare the impacts of the two SRM methods, we also made use of the corresponding outputs from piControl, abrupt4×CO2, and rcp45. We exclude the first decade following the large increase in forcing in common with other authors (Schmidt et al., 2012; Curry et al.

2014), and base our analysis on 40 years of data. All G1 and abrupt4×CO2 simulations are analyzed over a common period of simulation years 11 to 50, and the G4 and rcp45 simulations are analyzed from year 2030 to 2069. Equal weight is given to each model in the analysis, and climate extreme indices are calculated for each model before multi-model ensemble averaging is done.

**2.2 Climate extreme indices**

Here we use the climate indices defined by the Expert Team on Climate Change Detection and Indices (ETCCDI) (Zhang et al., 2011) to provide a comprehensive overview of temperature and precipitation

changes based on daily output of multi-models in GeoMIP and the Climate Model Intercomparison Project Phase 5 (CMIP5; Taylor et al., 2012). These indices have been widely used previously, both for observed weather (Donat et al., 2013) and model output (Tebaldi et al., 2006; Orlowsky and Seneviratne, 2012; Seneviratne et al., 2012) with Curry et al. (2014) using them for G1, Aswathy et al (2015) for G3, and Sillmann et al. (2013b) for CMIP5 models running the RCP scenarios.

We use six indices to describe temperature and precipitation extremes (Table 2), based on the daily output of surface air temperature and precipitation (tasmin, tasmax, pr). TXx and TNn are the maximum daily maximum and minimum daily minimum, respectively, of 2-m air temperature. These are absolute indices, representing the hottest or coldest day of a year or a month. The duration indices CSDI and WSDI are the longest number of consecutive days below (exceeding) the 10th (90th) percentiles of daily minimum

(maximum) temperatures (Table 2) calculated from piControl and indicate the length of cold spells and warm spells. The precipitation index Rx5day, the maximum 5-day precipitation sum in a month or year, can be taken as a rough indicator of increased flood probability (Frich et al., 2002). CDD is the maximum number of consecutive dry days with precipitation < 1 mm in a year, and is often referred to as a drought indicator.

All model output fields were re-sampled to a median model grid resolution of 144×90 (2.5º longitude × 2º latitude), which corresponds to the grid of the GISS-E2-R model. Following Curry et al. (2014) we adopted a first-order conservative remapping algorithm for non-integer variables (TXx, TNn, and Rx5day), (Jones, 1999), and nearest-neighbour interpolation for integer variables (CSDI, WSDI, and CDD).

**2.3 Normalization methods**

There are large differences in forcing between the G1 solar dimming and G4 stratospheric aerosol injection geoengineering schemes. The mean and extreme climates under the two types of geoengineering are quite different as will be shown below. To aid the comparisons, we adopt the

following normalization methods to compare spatially relative effectivities between solar dimming and stratospheric aerosol injection.

The normalized global spatial effects of solar dimming or stratospheric aerosol injection are defined as the grid mean difference relative to the global mean difference:

$$< X^{geo} - X^{ref} > = \frac{\overline{X}_{grid}^{geo} - \overline{X}_{grid}^{ref}}{|\overline{X}_{global}^{geo} - \overline{X}_{global}^{ref}|}$$

where the operator $<>$ denotes the normalized grid value, X is TXx, TNn, Rx5day or other climate field, an overbar denotes the average of each grid cell or the global average, the absolute operator $||$ in the denominator of the right term preserves the sign of the geoengineering anomaly. The superscript "geo" represents geoengineering experiments of G1 solar dimming or G4 stratospheric aerosol injection, the

superscript "ref" represents the reference experiments of abrupt4×CO2 or rcp45.

To normalize zonal mean difference in the climate extreme indices relative to the global mean difference, we use a similar formula:

$$< X^{geo} - X^{ref} > = \frac{\overline{X}_{zonal}^{geo} - \overline{X}_{zonal}^{ref}}{|\overline{X}_{global}^{geo} - \overline{X}_{global}^{ref}|}$$

where the operator $<>$ denotes the normalized zonal mean, an overbar denotes the zonal or global

average, the absolute operator $||$ in the denominator of the right term preserves the sign of the geoengineering anomaly.

## 3 Results

When discussing changes in climate variables the choice of reference scenario is important, though somewhat arbitrary. Curry et al. (2014) chose piControl as the reference for their study of G1, but here

we choose abrupt4×CO2 as the reference for G1 and rcp45 as the reference for G4. Our motivation for doing this is that because a return to pre-industrial era is not proposed or even likely to be desirable given the enormous quantities of GHG that would need to be removed from the climate system, in reality we will have to choose between either a world with GHG forcing, or with GHG forcing plus geoengineering. Atmospheric $CO_2$ concentrations equal those in abrupt4×CO2 would be reached by about the year 2100

under business-as-usual scenarios like RCP8.5.

### 3.1 TOA net radiation

The forcing of the G1 solar dimming and G4 stratospheric aerosol injection experiments are quite different, there can be a difference in the mean and extreme climate responses. The multi-model ensemble mean net radiation flux at the top of atmosphere (TOA) is 2.76 $Wm^{-2}$ and 0.004 $Wm^{-2}$ for the

abrupt×4CO2 and G1 experiments, and 1.63 $Wm^{-2}$ and 1.27 $Wm^{-2}$ for the rcp45 and G4 experiments during their 40-year analysis periods. Therefore, the G1 solar dimming and G4 stratospheric aerosol injection exert a reduction of 2.76 $Wm^{-2}$ and 0.36 $Wm^{-2}$ for net radiation fluxes at TOA respectively. The differences of mean net radiation flux at TOA over land and ocean between two geoengineering experiments and their reference experiments are show in Table 3. Although the ratio between the global

temporally averaged net radiation flux reductions at TOA is a factor of ~7.7, the spatial distribution of net radiation flux changes for the G1 and G4 ensemble means are quite similar, especially the positive TOA net radiation over Greenland, Antarctica, North Africa and West Asia, and the negative TOA net radiation over North America, Central Europe and tropical ocean basins (Figure 1). The entire ensemble shows a large and consistent positive TOA net radiation east of Greenland in the North Atlantic under

G1 solar dimming (Figure 1a), the region associated with the overturning part of the Atlantic meridional circulation (AMOC), and which under the G1 forcing was shown to be strongly affected by changes in radiative forcing and air/ocean heat exchange (Hong et al., 2017). However, differences are clearer when we investigate the spatial pattern of normalized effects exerted by the two SRM experiments, although most regions have differences close to zero for normalized solar dimming and stratospheric aerosol

geoengineering effects on TOA net radiation (Figure 1c). The G4 stratospheric aerosol injection geoengineering introduces a more effective reduction in TOA net radiation over the Northern Hemisphere, especially over the high-latitude continents, such as northern North America, Siberia and some regions of western Europe. The G1 solar dimming geoengineering introduces a more effective reduction in TOA net radiation over North Africa, northern South America, the Indian Ocean and tropical

Western Pacific. In contrast, many other equatorial regions, the Southern Ocean and the Intertropical and South Pacific Convergence Zones display small differences between normalized solar dimming and stratospheric aerosol injection effects.

The models show more consistent responses under G1 solar dimming then under G4 stratospheric aerosol injection, which is probably due to smaller signal-to-noise ratios under G4. The models are inconsistent

under both G1 and G4 over the Southern Ocean around Antarctica, where CMIP5 models also show large uncertainties in cloud radiative effects (Stocker et al. 2013). These results suggest that solar dimming and stratospheric aerosol injection geoengineering forcing may affect clouds differently in some models: low level clouds are important for radiative surface fluxes in the North Atlantic where differences

between G1 and G4 are positive, while higher clouds are more important in the deep tropical convection regions where differences are weakly negative. It is also possible that the different mean climate states between G1 and G4, and surface albedo changes due to sea ice and snow cover are responsible for the large differences in net radiation flux in the coastal Antarctic seas, and the more modest differences seen in the North Atlantic and Barents Sea along with Alaska and eastern Siberia.

In addition to different mean climate states and cloud responses, there are numerous sources of inter-model differences in response to solar dimming and stratospheric aerosol injection geoengineering. The G1 solar dimming assumes global uniform solar reduction, while under G4 sulphate aerosols are handled differently among the participating models. GISS-E2-R and HadGEM2-ES adopt stratospheric aerosol schemes to simulate the sulphate aerosol optical depth (AOD), BNU-ESM and MIROC-ESM use the

prescribed meridional distribution of AOD recommended by the GeoMIP protocol, CanESM2 specifies the uniform sulphate AOD (Kashimura et al., 2017). NorESM1-M specifies the AOD and effective radius which were calculated in previous simulations with the aerosol microphysical model ECHAM5-HAM (Niemeier et al., 2011; Niemeier and Timmreck, 2015). Although a prescribed AOD can be set, difference in assumed particle size for the stratospheric sulphate aerosols (Pierce et al., 2010) and the warming

effects of stratospheric aerosol (Pitari et al., 2014) cause difference in the SRM forcing.

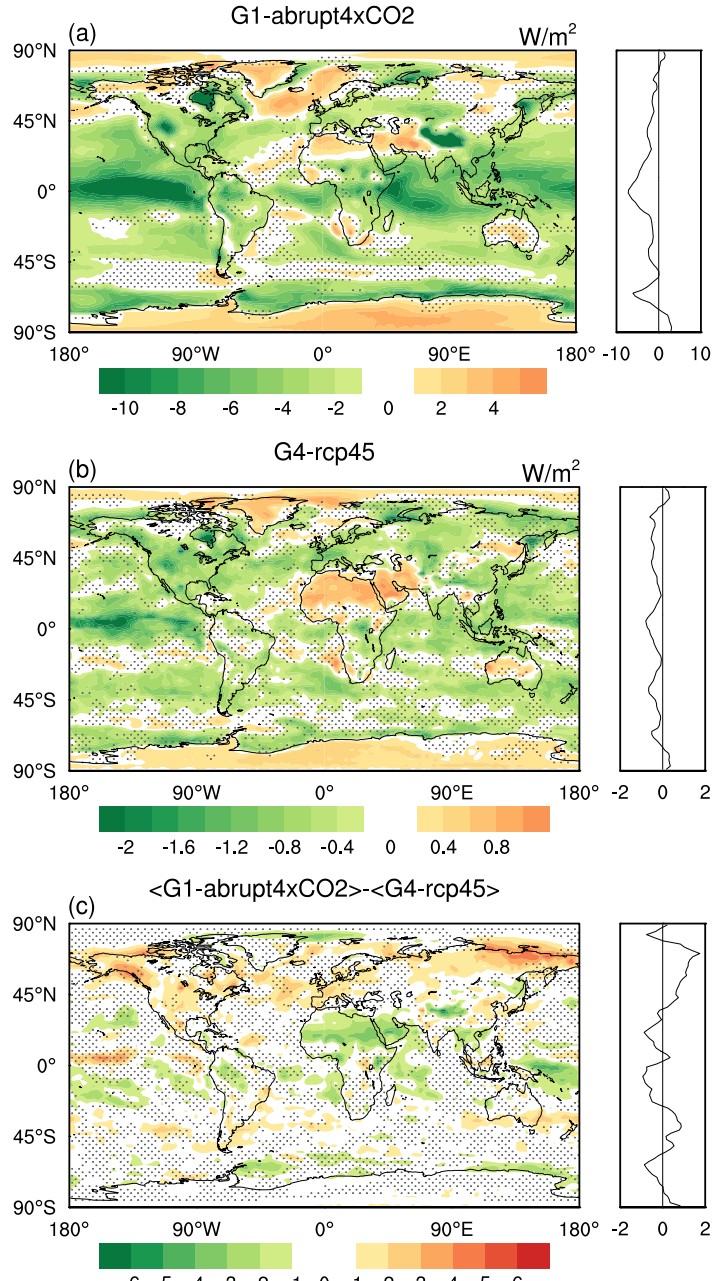

**Figure 1: Geographical distributions over the 40-year analysis periods of differences in net radiation flux at TOA between G1-abrupt4×CO2 (top), G4-rcp45 (middle). The bottom panel shows the differences in net radiation flux at TOA between normalized G1-abrupt4×CO2 and G4-rcp45. Stippling indicates regions where fewer than 5 of 6 models agree on the sign of the model response. The right sub-panels show the zonal average of the left sub-panels. Note that all three panels have different colour scales.**

**3.2 Probability distributions of monthly temperature and precipitation**

The G1 solar dimming and G4 stratospheric aerosol injection geoengineering greatly affect the mean climate states. The annual mean surface air temperatures are 291.0 K and 286.7 K for abrupt4×CO2 and

G1 experiments, 288.8 K and 288.3 K for rcp45 and G4 experiments respectively during their 40-year analysis periods. The global hydrological cycle strength is likewise reduced; the annual mean precipitation totals are 1125.8 mm and 1026.9 mm for abrupt4×CO2 and G1 experiments, 1098.4 mm and 1084.3 mm for rcp45 and G4 experiments (Table 3).

We computed the probability density functions (PDFs) of temperature and precipitation for each model

and average all models thereafter to get a general idea of the changes in the two geoengineering experiments (G1 and G4) compared to their baseline experiments (abrupt4×CO2 and rcp45). We first calculated the standardized monthly anomalies of monthly mean surface temperature in abrupt4×CO2 and rcp45 at every grid point in each model, i.e.

$$\tau_m^{ref} = (T_m^{ref} - \overline{T}_m^{ref})/\sigma_{T_m}^{ref}$$

where an overbar denotes the means of each month of the year calculated for the 11th to 50th years of the simulations and $\sigma_{T_m}^{ref}$ is the similarly calculated standard deviation for month m in the reference experiment, abrupt4×CO2 or rcp45. Next, we computed the monthly anomalies in G1 and G4 relative to the reference mean and standard deviation, i.e.

$$\tau_m^{geo} = (T_m^{geo} - \overline{T}_m^{ref})/\sigma_{T_m}^{ref}$$

The same algorithm was used to generate PDFs of precipitation. The multi-model mean PDFs use equal weights for each model. The results are shown in Figure 2. The PDFs for G1 and abrupt4×CO2 differ from those presented by Curry et al. (2014) as expected, due to the different choice of reference simulation. Curry et al. (2014) use the piControl as the reference experiment and compare the PDFs of G1 with piControl, which suggests temperature and precipitation perturbations that occur under

abrupt4×CO2 are all reduced to near-piControl values by G1 solar dimming geoengineering. In our study, we choose abrupt4×CO2 as the reference for G1, and rcp45 as the reference for G4, as we aim to investigate how the global mean and extreme temperatures and precipitation events may be ameliorated by G1 solar dimming and G4 stratospheric aerosol injection geoengineering compared to global warming.

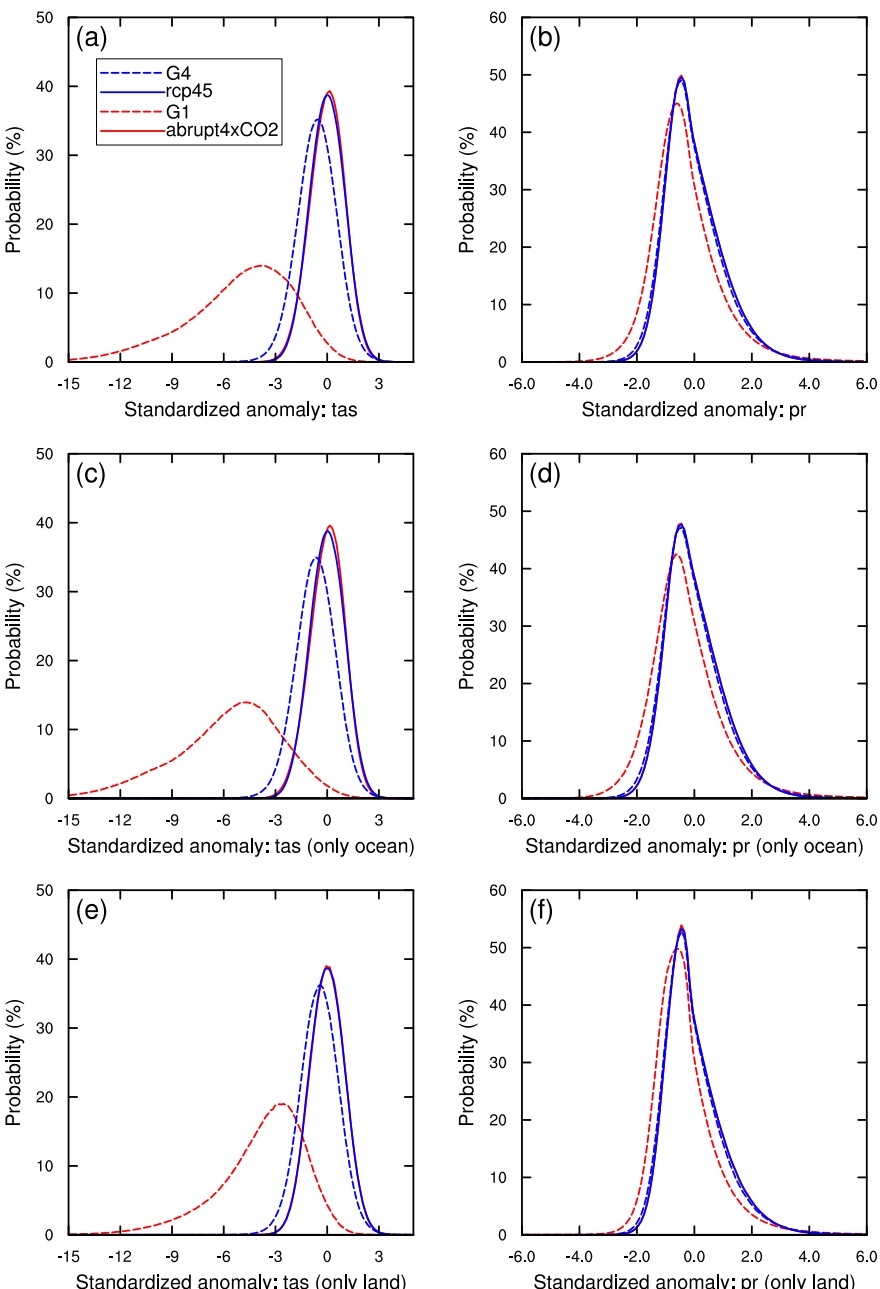

**Figure 2: Probability density distributions, normalized to 100%, of standardized monthly mean anomalies for the model ensemble average for four experiments: abrupt4×CO2 (solid red line), G1 (dashed red line), rcp45 (solid blue line), and G4 (dashed blue line). The PDFs of surface air temperature are shown in the left-hand panels, precipitation in the right-hand panels. Upper panels show global results, middle panels ocean-only, and lower panels land-only. *tas* denotes surface air temperature, while *pr* denotes precipitation.**

The PDFs of global temperature (i.e., including all points on each model grid; Fig. 2a) show a dramatic negative shift in G1 experiment, indicating cooling at nearly all locations in the models compared with abrupt4×CO2. Under G4 the PDFs display discernible differences from rcp45, mainly as negative anomalies—but the change is much smaller in G4-rcp45 than in G1-abrupt4×CO2. The relationships remain the same over the ocean and land domains as in the global. Figures 2c and 2e reveal that differences between temperature extremes over ocean and land domains are small, but the PDFs are more strongly centrally peaked over land than over ocean.

The PDFs of monthly precipitation display smaller differences between the two experiments than for temperature (right-hand panels of Fig.2). The PDFs are positively skewed in all cases, a general characteristic of precipitation and other positive definite climate variables (e.g., wind speed). The largest difference between G1 and abrupt4×CO2 occurs over ocean, where low tails are shifted towards more negative precipitation anomalies in G1 (Fig. 2f). As in the case of temperature, changes under G4-rcp45 are much smaller than under G1-abrupt4×CO2 with only a slight negative shift. Fig. 2d & f show that there are almost no differences between G4 and rcp45 over both land and ocean.

## 3.3 Global mean time series

Figure 3 shows the differences ($\Delta$) G1-abrupt4×CO2 and G4-rcp45 for TXx and TNn. In G1-abrupt4×CO2, $\Delta$TNn is significantly negative (Fig. 3a), with a multimodel mean value of $-5.1\pm0.4$ °C (one standard deviation, Table 3) over the 40 years analysis period (shaded region in Fig. 3a). By contrast, the extreme temperature index TXx has a smaller decrease with mean differences of $-4.4\pm0.3$ °C. Multi-model mean values of $\Delta$TNn are consistently a factor of ~1.2 more negative than those of $\Delta$TXx (Fig. 3a and c), indicating a much stronger response of night-time low temperatures to a reduction in the solar constant, relative to daytime high temperatures. This is also the case in G4-rcp45, but with much smaller magnitude ($\Delta$TNn = $-0.7\pm0.1$ °C and $\Delta$TXx = $-0.6\pm0.1$ °C), Figs. 3b, d. The larger change in TNn relative to TXx was also found in the GeoMIP G1-piControl simulations analyzed by Curry et al. (2014), and in the increasing GHG scenarios in CMIP3 as well as CMIP5 (Tebaldi et al., 2006; Orlowsky and Seneviratne, 2012; Sillmann et al., 2013a). The explanation for the difference in daytime and night-time response is due to much stronger response of night-time low temperatures than daytime high temperatures. TNn is reduced more than TXx under G1 (and G4) because of the reduced warming under geoengineering, lower temperatures and reduced longwave effects throughout the whole day and night,

although the reduced shortwave surface heating impacts daytime temperatures directly under G1 (and

G4). The GISS-E2-R model has a noticeably weaker response measured by ΔTNn and ΔTXx changes

than the other models. This is due to its relatively weak warming under abrupt4×CO2 as shown by Curry

et al. (2014), meaning that the degree of solar dimming needed by G1 SRM is also weaker than for other

models. The changes in radiative forcing at both short and long wavelengths are thus smaller in GISS-

E2-R and the changes in various climate indicators are also smaller (Yu et al., 2015).

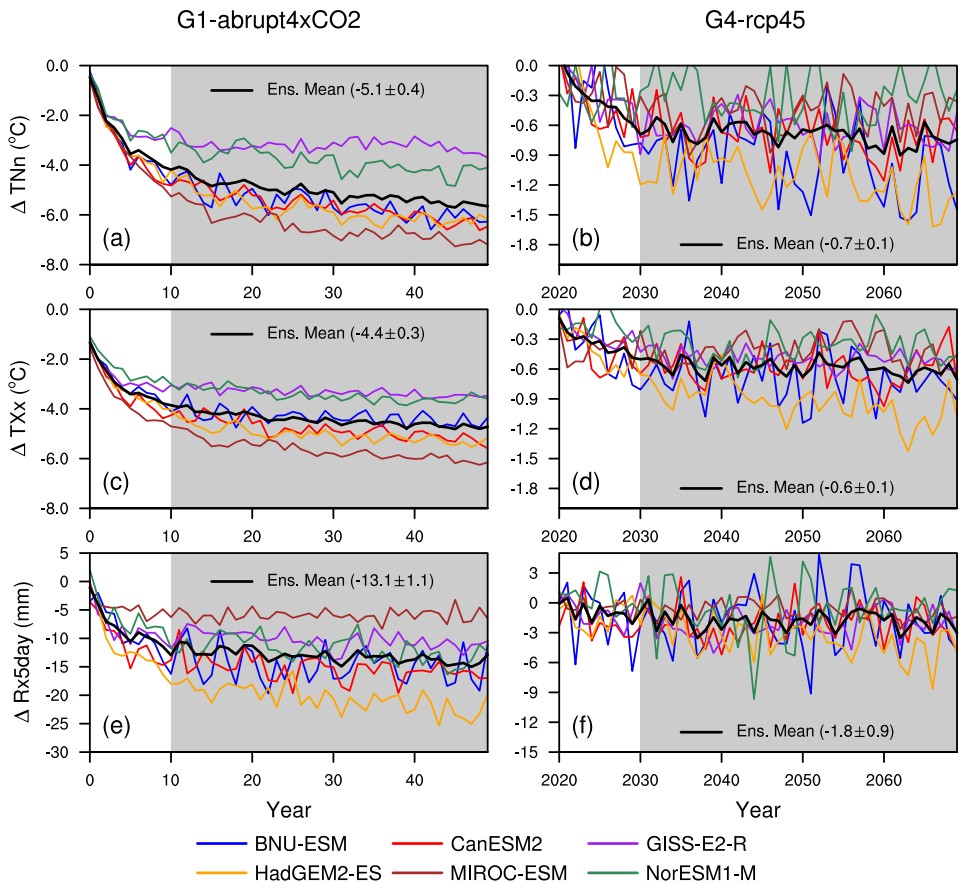

**Figure 3: Time series of the difference of global mean extreme indices (as labelled on each left-hand panel's**
**ordinate) between G1 - abrupt4×CO2 (left column) and G4 - rcp45 (right column) for all models analyzed.**
**The black curves are the multimodel means, and gray shading indicates the 40-year analysis period for each**
**experiment used in this study, with the ensemble mean value also shown on each panel.**

The corresponding result for the extreme precipitation index, Rx5day is a significant reduction under G1

(Fig. 3e) with a multi-model mean value of -13.1±1.1 mm, indicating an overall weakening of the

hydrological cycle. This feature was noted for non-extreme indices in the G1 experiments analyzed by

Schmidt et al. (2012) and Kravitz et al. (2013b). In contrast, the index for G4-rcp45 (Fig. 3f) is near-zero, though slightly negative on the whole, with the multi-model mean value of -1.8±0.9 mm. The partly positive Rx5day for G4-rcp45 reflects the climate variability simulated by models and lower signal-to-noise ratio. The mean temperature difference under G1 solar dimming is -4.3 °C, and -0.5 °C under G4

stratospheric aerosol, hence a ratio of 8.1, larger than extreme aspects of temperature: 7.3 for TNn, and 7.6 for TXx (Table 3). The corresponding ratio for mean precipitation is 7.0, whereas extreme precipitation indicated by Rx5day has a ratio of 7.3., similar to TNn and TXx. In general, G1 solar dimming and G4 stratospheric aerosol injection seem equally effective at changing extreme precipitation as well as extreme high and low temperatures, though solar dimming seems more effective than

stratospheric aerosol injection at controlling mean temperature.

If relative humidity and atmospheric circulation remain relatively unchanged, then intense precipitation amount is governed by total precipitable water in the atmosphere, which the Clausius–Clapeyron relation says scales with mean temperatures (Allen and Ingram, 2002). The global mean precipitation decreases 2.1±0.4% per Kelvin in response to G1 solar dimming, and 2.7±1.0% per Kelvin in response to G4

stratospheric aerosol injection. The GISS-E2-R model contributes a relatively large portion to the spread of scaling between mean precipitation and temperature with a value of 4.5% per Kelvin for G4. If excluding the GISS-E2-R model, the global mean precipitation decreases 2.0±0.4% per Kelvin in response to G1 solar dimming, and 2.3±0.5% per Kelvin in response to G4 stratospheric aerosol injection. The scaling between mean precipitation and mean temperature under G1 and G4 is smaller than 3.4%

precipitation change per Kelvin estimated from other coupled models under long-term equilibrium climate in response to doubling $CO_2$ (Allen and Ingram, 2002). The global mean Rx5day decreases 3.4±1.0% per Kelvin in response to G1 solar dimming, and 4.3±2.6% per Kelvin in response to G4 stratospheric aerosol injection. GISS-E2-R gives global mean Rx5day decreases 9.5% per Kelvin for G4. If excluding GISS-E2-R model, the global mean Rx5day decreases 3.4±1.1% per Kelvin in response to

G1 solar dimming, and 3.3±0.6% per Kelvin in response to G4 stratospheric aerosol injection. The scaling of mean precipitation and mean temperature is expected to be much less than the 6.5% per Kelvin implied by the Clausius–Clapeyron relation, as the global-mean precipitation is primarily constrained by the availability of energy not moisture (Pall et al., 2007). The scaling of Rx5day and mean temperature under G1 and G4 is close to, but still weaker than the Clausius–Clapeyron relation, probably because

Rx5day is not really an index of the heaviest rainfall events that are expected to be constrained by the

Clausius–Clapeyron relation. The Clausius–Clapeyron relation implies the same scaling of extreme precipitation and mean temperatures under both G1 and G4 experiments, which is the case here for five of six models, but not the GISS-E2-R model.

### 3.4 Spatial Response in Extremes

Geographical patterns of difference between the two SRM scenarios: i.e., G1- abrupt4×CO2 and G4-rcp45 are shown in Figure 4.

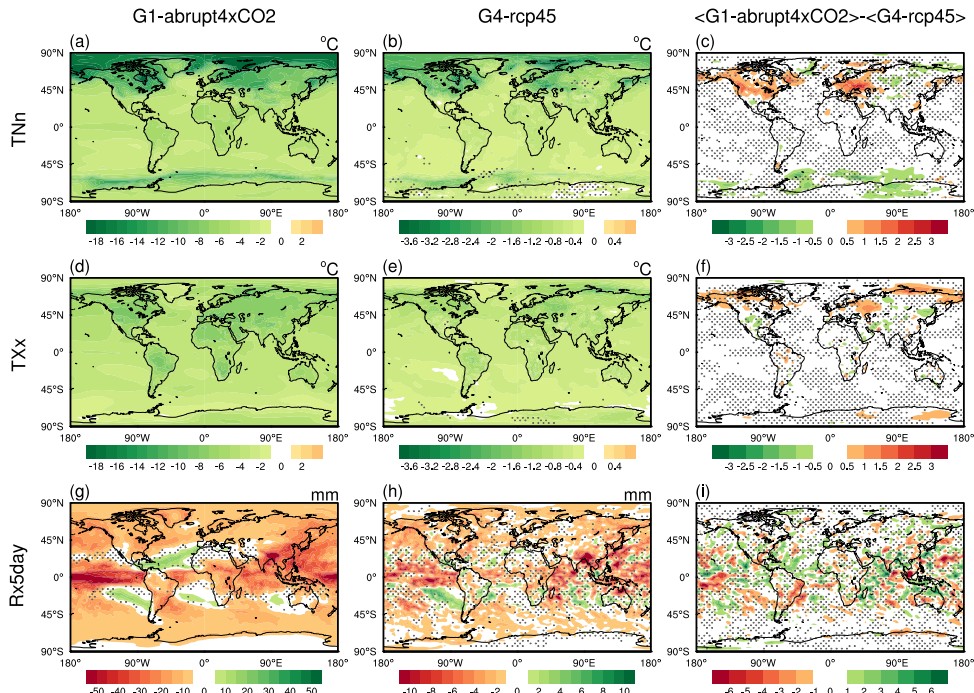

**Figure 4: Geographical distributions over the 40-year analysis periods of the differences G1 - abrupt4×CO2 (left column), G4 - rcp45 (middle column), and differences between normalized G1 - abrupt4×CO2 and G4 -**

**rcp45 (right column) for the extreme indices TNn (top row), TXx (middle row), and Rx5day (bottom row). Stippling indicates regions where fewer than 5 of 6 models agree on the sign of the model response. Note that panels have different colour scales.**

The cooling patterns seen for TNn (Fig. 4a,b) are similar but with a larger signal for G1-abrupt4×CO2 than G4-rcp45, with the signature of polar amplification evident in both hemispheres but primarily in the

Arctic. Several studies have considered the reasons behind this effect. Similar patterns occur in simulations of mean temperatures under both GHG warming scenarios and under geoengineering scenarios (Schmidt et al., 2012; Curry et al., 2014; Kravitz et al., 2013b). Pithan and Mauritsen (2014) conclude that in climate models it is primarily the temperature feedback with surface albedo of secondary

importance in producing Arctic amplification under GHG forcing. While James et al. (2010) concluded that changes in sea ice cover play a leading role in recent Arctic temperature amplification for GHG forcing. The spatial pattern under geoengineering is due to the seasonal differences in longwave and shortwave forcing. Tilmes et al., (2014) and Hong et al., (2017) note the importance in poleward heat

transport by reduction in the strength of the meridional overturning circulation under GHG forcing. Geoengineering has been shown to mitigate sea ice loss (Moore et al., 2014; Berdahl et al., 2014), and also reduce the decline in ocean poleward heat transport (Hong et al., 2017) relative to GHG forcing, but these changes do not completely counter the increase in radiative flux due to GHG forcing. In addition to the cooling patterns seen in the Arctic, TNn presents a cooling in the ocean around the Antarctica,

which is not seen in TXx.

A notable feature in Fig. 4a, 4b, 4d and 4e is the larger cooling of TNn and TXx over land compared with oceans, also expressed in Table 3. The land-sea cooling contrast is larger for TXx than TNn (Fig 4d, e; Table 3), and TXx shows more uniform cooling than TNn across all latitudes. This feature is consistent with the stronger relationship of shortwave forcing to TXx. Under GHG warming scenarios, heat capacity

differences, contrasts in surface sensible and latent fluxes, and boundary layer differences lead to contrasts opposite to those under G1 and G4 (Sutton et al., 2007; Joshi et al., 2008). Under G1 and G4, GHG warming occurs 24 hours a day, while reduced solar radiation is more effective in reducing day-time temperatures (TXx), with the land-sea heat capacity differences further enhancing TXx over TNx. The land–sea cooling effects under G4-rcp45 (Fig. 4b,4e) are consistent with Volodin et al. (2011) who

found increased land–sea cooling contrast in annual mean temperature using the INMCM model forced with 4 Mt S/year equatorial stratospheric aerosol injection.

Comparing Figs. 4a,b and d,e shows that the magnitude of $\Delta$TNn is larger than that of $\Delta$TXx at high latitudes. The strongest cooling in TXx of up to -9.9°C under G1-abrupt4×CO2 (Fig. 4d) generally occurs in the interior of the continents as previously discussed, such as in South and North America, Eastern

Europe, north-central Eurasia and Australia. The pattern is similar in G4-rcp45 but with a smaller magnitude. Fig. 4c, f show the differences between the normalized changes G1-abrupt4×CO2 and G4-rcp45. The stratospheric aerosol injection more effectively reduces the TNn in northern North America and western Europe compared with solar dimming, while the solar dimming more effectively reduces TNn in the Siberian coastal region, Eastern Antarctica and the adjacent ocean regions. The stratospheric

aerosol also effectively reduces the TXx in northern North America and central Europe compared with

solar dimming, but with a smaller spatial extent and magnitude compared with TNn. Stratospheric aerosol is more effective at reducing TXx in the Siberian coastal region, while the solar dimming seems more effective on reducing TNn there. Averaged over the globe, the magnitude of the extreme temperature anomalies under G1-abrupt4×CO2 is a factor of ~8 larger than under G4-rcp45, simply due to the much

larger forcings in G1 relative to G4 (Table 3). Significantly smaller ratios for ΔTNn occur in central North America, eastern China and the northern Mediterranean as well as areas in Antarctica, with significantly larger ratios mainly in the Southern Ocean (not shown). Corresponding results for ΔTXx show smaller ratios in northern North America and Asia, West Asia, as well as areas in Antarctica, with larger ratios mainly in northeastern China and southern North America as well as in some ocean areas.

Using geoengineering to alleviate surface warming from increasing GHGs concentrations decreases global-mean precipitation (Schmidt et al., 2012; Kravitz et al., 2013b) as well as the wettest five days index (Rx5day), representing an extreme aspect of the precipitation distribution (Curry et al., 2014). The ensemble means show that Rx5day is strongly reduced over equatorial regions, especially in the equatorial Pacific and southern flank of the Tibetan Plateau (Fig. 4g,h). This is due to increased

atmospheric stability and suppression of convection under geoengineering (Bala et al.,2008). Fig. 4g,h and Curry et al. (2014) show some robust increases in the tropics, northwest Africa, the Mediterranean Sea and areas of the subtropical oceans, which consistently display decreased Rx5day under abrupt4×CO2 compared to G1. This has been attributed to a weaker Hadley cell due to weaker radiative forcing (Tilmes et al., 2009), but more recent analysis of the tropical circulation suggests more complex

interactions between radiative forcing and Hadley cell extent and intensity. Under GeoMIP G1 experiment, the Hadley cell edges remain at their preindustrial width latitudinally, despite the residual stratospheric cooling associated with elevated carbon dioxide levels (Davis et al., 2016; Guo et al., 2018). The damping of the seasonal migration of the Intertropical Convergence Zone (ITCZ) within the Hadley cell under G1 is associated with preferential cooling of the summer hemisphere (Smyth et al., 2017).

The spatial pattern for G4-rcp45 is not as coherent as that for G1-abrupt4×CO2, although Rx5day also increases mainly in the subtropics and decreases at equatorial regions, high latitudes and over most land areas (Fig. 4h). The noisy G4-rcp45 response is also seen in the climatological mean precipitation (Yu et al.,2014) under G3 and G4, as well as in the consecutive dry days (CDD) index under the G3 experiment (Aswathy et al., 2015). Furthermore, monsoonal regions including East Asia and India exhibit a reduction

in Rx5day under G1-abrupt4×CO2, which may be attributed to a weakened monsoon. Tilmes et al. (2013)

observed, using a larger ensemble of models, that G1-abrupt4×CO2 results in a robust decrease in monsoonal precipitation, while it increases under abrupt4×CO2. Reduced Rx5day over monsoon regions is an indicator of weakened monsoon (Fig. 4g), because although the extreme precipitation index is calculated on an annual basis, it is dominated by wet season precipitation, particularly in monsoon areas (Klein Tank et al., 2006). However, the change under G4-rcp45 is not as robust (Fig. 4h), due at least partially to lower mean temperature changes and land-sea thermal contrast, and therefore smaller signal-to-noise ratios compared with G1-abrupt4×CO2. The difference between normalized change of G1-abrupt4×CO2 and G4-rcp45 is noisy and without coherent patterns (Fig. 4i).

As the tropical extreme precipitation change constitutes a large percentage of global extreme precipitation change in response to two type geoengineering schemes (Fig. 4g, 4h), it is interesting to know how the G1 solar dimming and G4 stratospheric aerosol injection affect major rain types in tropical regions. We compared tropical ($\pm$ 30º lat.) relative frequency changes of four major daily rain types: light rain (<0.3 mm/day), moderate rain (0.9–2.4mm/day), heavy rain (>9mm/day) and an extremely heavy rain type (>24 mm/day) according to daily rain types used in Lau et al. (2013). All six models show consistent shift in rain regime, with a decrease in the frequency of extremely heavy rain by -22.3% for G1 and -3.6% for G4, heavy rain by -5.2% for G1 and -0.6% for G4, and consistent increase in the frequency of light rain by +4.4% for G1 and 0.5% for G4.

**3.5 Extreme Duration Response**

The TXx, TNn and Rx5day indices discussed above all characterize aspects of the absolute magnitude of climate extremes. We now analyze the duration indices shown in Figure 5: cold spell duration (CSDI), warm spell duration (WSDI), and consecutive dry days (CDD).

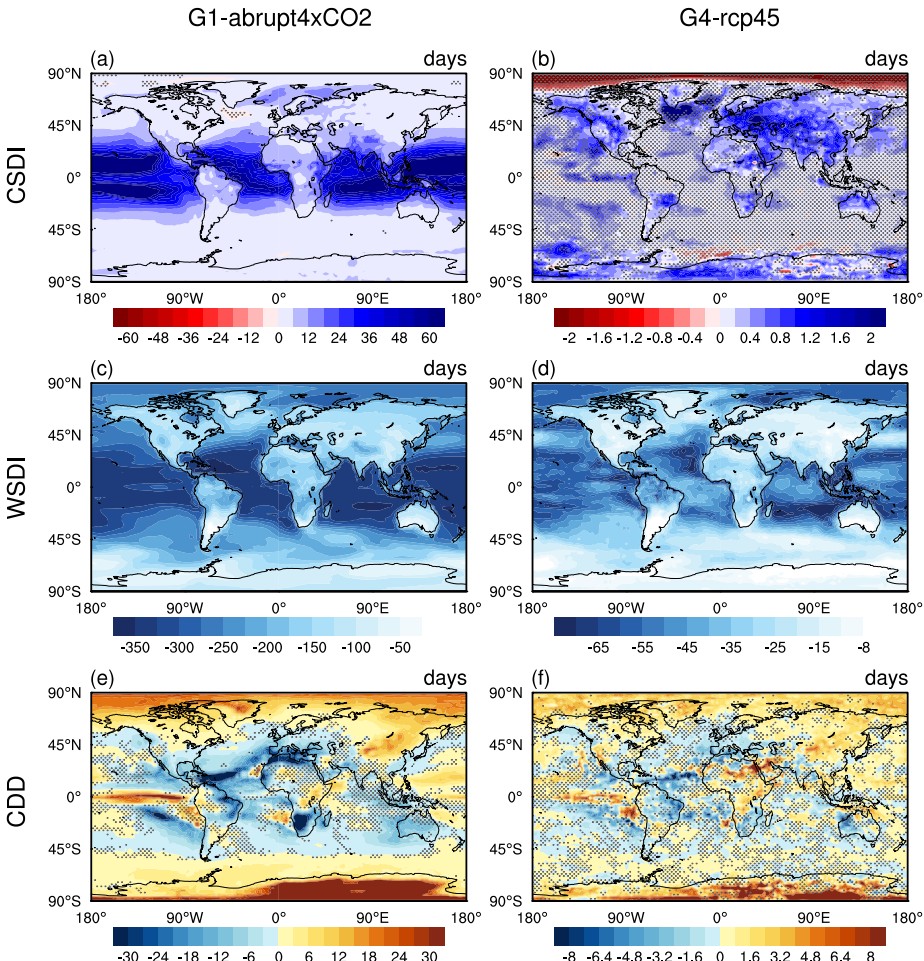

**Figure 5: Geographical distributions of differences, G1 - abrupt4×CO2 (left column) and G4 - rcp45 (right column) for the extreme duration indices (a, b) CSDI, (c, d) WSDI, and (e, f) CDD, taken over the 40-year analysis periods. Stippling indicates regions where fewer than 5 of 6 models agree on the sign of the model response.**

CSDI increases worldwide in the G1-abrupt4×CO2 anomaly (Fig. 5a), due to the strong negative shift of the PDF of surface temperature for G1 relative to abrupt4×CO2 (Fig. 1a). The most striking feature of Fig. 5a is the robust increase in CSDI over the tropical oceans with ΔCSDI exceeding 50 days per year over large regions, indicating that the region is sensitive to reduced solar radiation. Most of the CSDI differences over land in G1-abrupt4×CO2 are robust, with the notable exception of tropical regions such as India and Indonesia, which experience an increase in cold spell duration of more than 30 days (Fig. 5a). In contrast to the large response under G1-abrupt4×CO2, the pattern in G4-rcp45 is incoherent with

wide disagreement about the sign of change between the models except for a robust increase over the continental regions of Eurasia and North America.

The spatial pattern of WSDI (Fig. 5c) shows a notable decrease over the tropical oceans, exceeding 300 days per year. The pattern is similar to CSDI but of larger magnitude and with a more widespread decrease over land areas such as eastern south America and the Tibetan Plateau (Fig. 5c). Comparison of Figs. 5a and 5c shows that in G1-abrupt4×CO2, WSDI decrease much more strongly over the tropical and subtropical oceans than do CSDI. The pattern of WSDI in G4-rcp45 is similar to that in G1-abrupt4×CO2, except in the equatorial ocean regions, which is also noticeable in the pattern of changes in CSDI (Figs. 5a, b).

The relatively small, but robust, changes in annual extreme temperature in the tropics apparently contradict the rather large robust increases in CSDI and decreases in WSDI (Fig. 5a, c) under solar dimming, but were also reported by Curry et al (2014). Cold spell and warm spell duration are related to the magnitude of changes in mean temperature relative to the short-term temperature variability. They are sensitive to the underlying climatological temperature variability of the respective region (Radinovio et al., 2012), which is small in the tropics and larger in the extra-tropics. A small shift in mean temperature can lead to large changes in the duration of cold and warm spells, which may have relatively large impacts on ecosystems (Corlett, 2011). The more robust results (lack of stippling in Fig. 5d) for the WSDI anomalies under G4 than for the CSDI are due to the significant cooling imparted by G4 relative to rcp45, as reflected by the color bar ranges. For example, BNU-ESM shows small increases in CSDI over the Arctic Ocean, while HadGEM2-ES shows strong decreases and other models have spatially varying results in G4 relative to rcp45. This may be due to Arctic amplification linked to, among other things discussed in Section 3.3, loss of sea ice, which occurs under both rcp45 and G4 simulations (Berdahl et al., 2014). There is a wide model spread in model-projected Arctic sea ice extent, although HadGEM2-ES and BNU-ESM produce similar sea ice patterns while MIROC-ESM simulates essentially no ice cover in autumn (Berdahl et al., 2014). The spatial pattern of the TOA net radiation flux varies relatively more in G4-rcp45, ranging from -0.22 to -0.56 $Wm^{-2}$, while comparatively ranging from -2.29 to -3.40 $Wm^{-2}$ in G1-abrupt4×CO2 (Figure A1, A2 in Appendix). As simulation of sulphate aerosols differs among the participating G4 models, the spatially varying forcing results in very different cooling patterns particularly at high latitudes.

The equatorial Pacific in the vicinity of the ITCZ displays increases in CDD under G1-abrupt4×CO2 at

the same locations (Fig. 5e) as Rx5day decreases (Fig. 4g). This may be related to the reduced latitudinal extent of seasonal movement of the ITCZ under G1 as noted in previous studies (Schmidt et al., 2012; Smyth et al., 2017). Anti-correlation between CDD and Rx5day can also be seen for decreases in CDD and increases in Rx5day in the tropical Atlantic, South Atlantic and the southeast Pacific dry zone. Both

northern and southern high latitudes, and large parts of Eurasia display increases in CDD and decreases in Rx5day (Fig. 5e, 4g). CDD decreases in the desert regions of northern Africa, southwestern Africa, Australia and southwestern North America, which are strongly influenced by the descending branch of the tropical Hadley cell. This implies most places have fewer droughts under the geoengineering simulation than without it. Fig. 5f shows that the pattern in G4-rcp45 is similar to G1-abrupt4×CO2 but

noisier.

### 3.6 Seasonality and zonal mean changes

We now examine the zonal structure and seasonality of changes in the climate extreme indices. Seasonal analysis is performed only for indices that can be presented on a monthly basis, i.e., TXx, TNn, and Rx5day. There are large temperature differences between G1 and abrupt4×CO2 simulations over polar

regions due to residual polar amplification effects, and similarly for G4-rcp45 but with smaller magnitude.

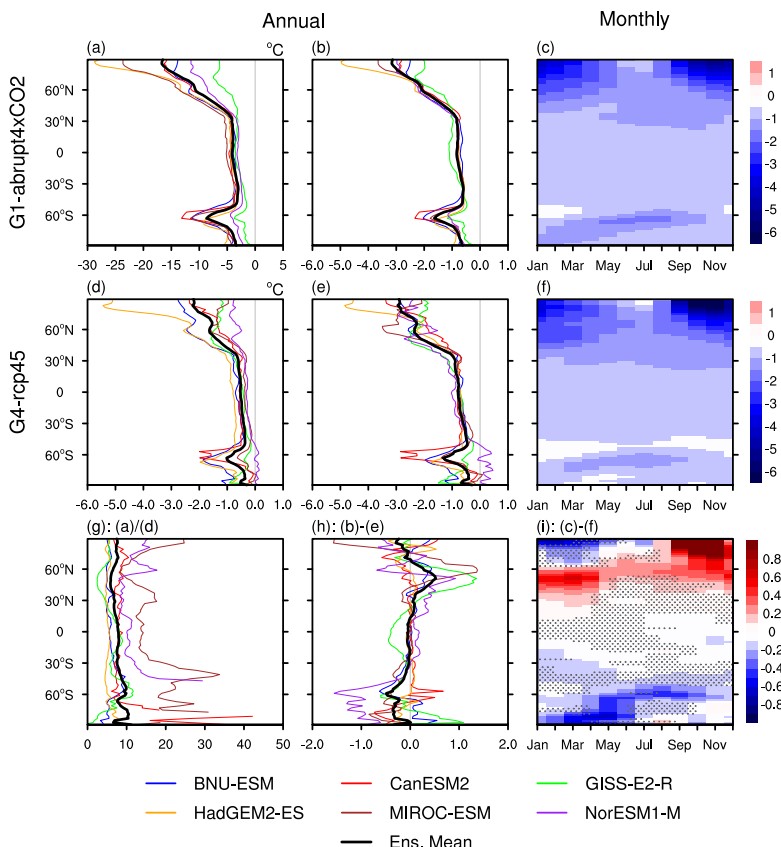

**Figure 6: Absolute difference of annual zonal mean in the extreme low temperature TNn (left column), Normalized difference with respect to annual zonal mean (middle column) and monthly zonal mean (right column) in TNn: (a), (b), (c) G1 – abrupt4×CO2, (d), (e), (f) G4 – rcp45, (g) the ratio between absolute G1 – abrupt4×CO2 and G4 – rcp45, (h) the annual zonal mean difference between normalized G1 – abrupt4×CO2 and G4 – rcp45, (i) the monthly zonal mean difference between normalized G1 – abrupt4×CO2 and G4 – rcp45 taken over the 40-year analysis period. In panel (i) red colours indicate relatively greater changes with G4 and blue colours with G1, stippling indicates regions where fewer than 5 of 6 models agree on the sign of the model response. 3×3-point smoothing was applied to the seasonal-latitude change.**

The left panels in Figure 6 display the zonal and annual mean anomalies, ΔTNn. The response in G1 compared with abrupt4×CO2 is of course uniformly negative (Fig. 6a), with multi-model mean annual peak values of -17°C near 90°N and -8°C near 65°S. In G4-rcp45, most models simulate a much smaller negative response.

As shown in the right panels of Fig. 6, the Arctic (defined as the region north of 67.5°N) cooling of TNn has a distinct seasonal character under both G1-abrupt4×CO2 and G4-rcp45. Arctic amplification peaks (up to -25°C in G1-abrupt4×CO2, and -5°C in G4-rcp45, not shown) in early winter (November to December). In winter under abrupt4×CO2, the warm ocean forms only limited seasonal sea ice cover and

produces low cloud cover increasing downward longwave radiation and hence remains relatively warm. However, under G1 the sea ice cover is largely multi-year (Moore et al., 2014), hence is thicker and maintains a much lower surface temperature as the ice cover cools compared with open ocean. In summer, surface melting on the ice, which is still present in most models under abrupt4×CO2, and the large thermal inertia of the ocean tend to drive minimum surface temperatures under both G1 and abrupt4×CO2 close to the freezing point. A distinct TNn decrease is observed in the high latitudes of the Southern Ocean from April to October in both G1-abrupt4×CO2 and G4-rcp45, likely also due to sea ice processes. The annual zonal mean pattern of G4-rcp45 (Fig. 6d) is comparable to G1-abrupt4×CO2 (Fig. 6a), but weaker by a factor of 7 to 9 in terms of their absolute magnitudes (Fig. 6g). Fig. 6b, 6e show normalized zonal and annual mean anomalies of $\Delta$TNn. Although G1 and G4 possess different geoengineering radiative forcings, the normalized zonal and annual mean anomalies of $\Delta$TNn display similar patterns and magnitudes. The differences of normalized response in TNn in Fig. 6h is nearly spatially uniform, and close to zero in the annual mean, except for the high latitudes of the Northern and Southern Hemispheres. This is consistent with Fig. 6g, which shows the absolute ratio of response in TNn, and which implies that a constant scaling of the zonal and annual mean response to G4 would be close to that of G1. Hence, in Fig. 6i, values less than zero indicate where solar dimming is an intrinsically stronger geoengineering agent than stratospheric aerosols, and values above zero highlight where stratospheric aerosols tend to be more effective, with a value around zero meaning that solar dimming and stratospheric aerosols are equally effective. Fig. 6i shows that TNn in the northern high latitude springs and summers is affected much more by solar dimming than by stratospheric aerosol injection. A similar response is also present in the wintertime and springtime Southern Ocean. The only regions where stratospheric aerosol injection induces a significantly larger response than solar dimming is in the high Arctic in winter and latitudes between 40°-60°N in spring and winter, suggestive of a longwave radiative effect of the aerosols.

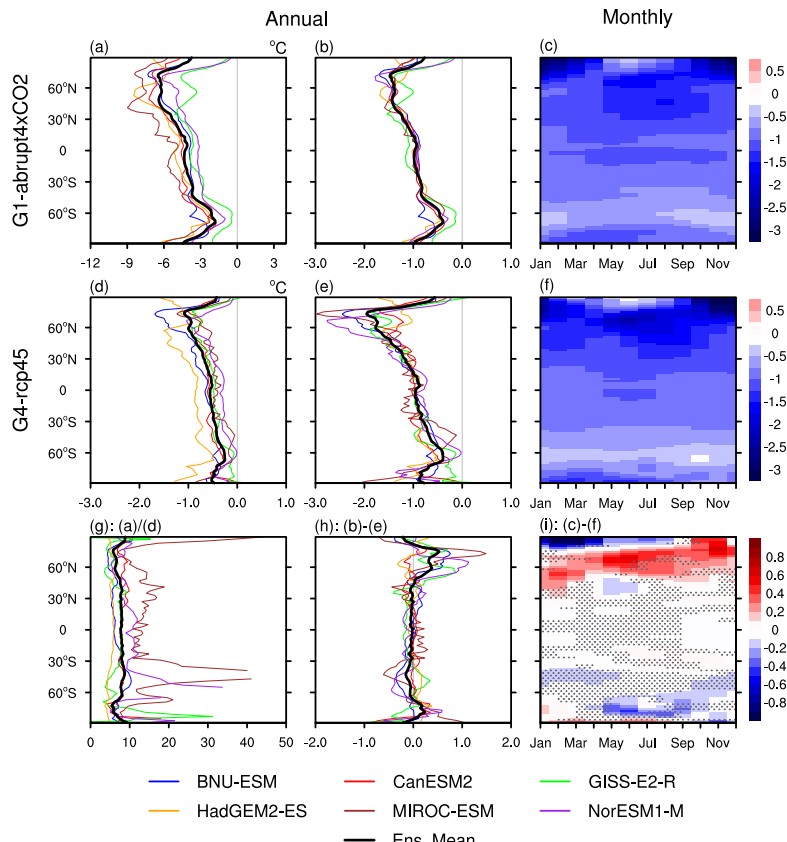

**Figure 7: Absolute difference of annual zonal mean in the extreme high temperature TXx (left column), Normalized difference with respect to annual zonal mean (middle column) and monthly zonal mean (right column) in TXx: (a), (b), (c) G1 – abrupt4×CO2, (d), (e), (f) G4 – rcp45, (g) the ratio between absolute G1 – abrupt4×CO2 and G4 – rcp45, (h) the annual zonal mean difference between normalized G1 – abrupt4×CO2 and G4 – rcp45, (i) the monthly zonal mean difference between normalized G1 – abrupt4×CO2 and G4 – rcp45 taken over the 40-year analysis period. In panel (i) red colours indicate relatively greater changes with G4 and blue colours with G1, stippling indicates regions where fewer than 5 of 6 models agree on the sign of the model response. 3×3-point smoothing was applied to the seasonal-latitude change.**

Figure 7 shows that the multi-model mean ΔTXx in both G1−abrupt4×CO2 and G4−rcp45 are of smaller magnitude than ΔTNn at high latitudes (Fig. 6). TXx is much less latitudinally variable than TNn both in G1-abrupt4×CO2 and G4-rcp45 (compare Figs. 6a, d and 7a, d). The signature of polar amplification (especially in the Northern Hemisphere) is evident in ΔTNn (Fig. 6a, d) whereas an asymmetric north – south response is evident for ΔTXx. The north-south ΔTXx asymmetry reflects the global land distribution, with ΔTXx more strongly affected over land than ocean (Fig. 4 d, e). The strongest cooling in G1-abrupt4×CO2 is found in Arctic winter, when more winter Atlantic cyclones track into the high Arctic under abrupt4×CO2 than G1 (Moore et al., 2014), and in the Northern mid-latitude summers,

consistent with the regions where snow-albedo feedback and the soil moisture effect are strongest (Orlowsky and Seneviratne, 2012; Seneviratne et al., 2006; Diffenbaugh et al., 2007). Geoengineering leads to increases in both snow cover and soil moisture which lowers surface sensible heat flux, raises heat capacity and thus lowers sensitivity of temperature to radiative forcing changes (Curry et al., 2014, Dagon and Schrag, 2017). Similar patterns hold for G4-rcp45. As with TNn in Fig. 6h, the differences of normalized response in TXx is remarkably spatially uniform and around zero (Fig. 7h). Fig. 7i suggests that the relative effectiveness of stratospheric aerosols and solar dimming is similar, except for the Arctic, and perhaps Antarctica, where aerosols appear more effective than dimming in winter. Since the lack of shortwave radiative forcing during winter would not lead to differences in solar dimming or aerosol response, atmospheric circulation changes are implicated. The tropical lower stratospheric radiative heating due to stratospheric aerosol would drive a thermal wind response, which would intensify the stratospheric polar vortices. In contrast, solar dimming does not produce this effect and so there is little intensification of the polar vortex in G1. Therefore, the response of the northern hemisphere polar vortex to solar dimming geoengineering is much weaker than under stratospheric aerosol injection (Ferraro et al. 2015). A strengthening of the wintertime stratospheric polar vortices occurs under G4, tending to cool polar surface temperatures, which is consistent with wintertime northern hemisphere TNn and TXx patterns shown in Fig. 6i and 7i. This also promotes heterogeneous reactions on aerosols depleting stratospheric ozone, further strengthening the stratospheric vortex and cooling the poles (Tilmes et al., 2009), although this effect is not included in the models used in this study.

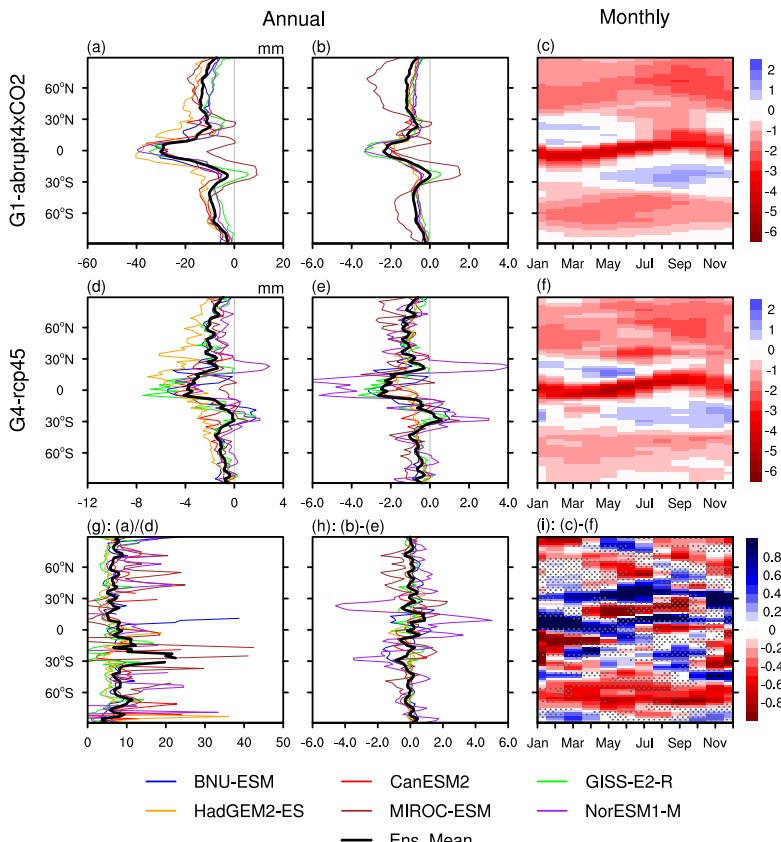

**Figure 8: Absolute difference of annual zonal mean in the extreme precipitation Rx5day (left column), Normalized difference with respect to annual zonal mean (middle column) and monthly zonal mean (right column) in Rx5day: (a), (b), (c) G1 – abrupt4×CO2, (d), (e), (f) G4 – rcp45, (g) the ratio between absolute G1 – abrupt4×CO2 and G4 – rcp45, (h) the annual zonal mean difference between normalized G1 – abrupt4×CO2 and G4 – rcp45, (i) the monthly zonal mean difference between normalized G1 – abrupt4×CO2 and G4 – rcp45 taken over the 40-year analysis period. In panel (i) blue colours indicate relatively greater changes with G4 and red colours with G1, stippling indicates regions where fewer than 5 of 6 models agree on the sign of the model response. 3×3-point smoothing was applied to the seasonal-latitude change.**

The results for the extreme precipitation index Rx5day are shown in Figure 8. Under GHG forcing alone, both observations and simulations show wet seasons get wetter and dry seasons get drier (Chou et al., 2013). The months July-November in the Northern Hemisphere and February-May in the Southern Hemisphere become somewhat drier (10-16%) under geoengineering. Fig. 8f displays a similar summer/winter, tropical wet/dry season effect for G4, where it appears over a slightly narrow latitude range and is slightly delayed relative to G1. Increased occurrence of extreme rainfall under G1 (>16%) is expected during winter and spring for the subtropical regions of both hemispheres. The effect on Rx5day is largest in April to November in the Southern Hemisphere, which roughly corresponds with

the subtropical wet season. The path of darker red in Figs 8c and 8f appear to follow quite closely the seasonal migration of the ITCZ which wanders near the sub-solar point. Smyth et al. (2017) report that the seasonal amplitude of migration of the ITCZ is reduced under G1 relative to piControl and this would be consistent with the seasonal reduction in Rx5day along the dark red paths in Figs 8c and 8f. Tilmes et

al. (2013) noted that in the G1 experiment precipitation in the tropics is reduced by around 5% with a larger interannual variability and spread among the models over land compared to the ocean. Furthermore there is considerable reduction in frequency of heavy precipitation ($> 8$ mm day$^{-1}$) over the tropics and at the same time an increase in the frequency of small and moderate precipitation intensity. This is consistent with the seasonal analysis shown in Fig. 8 if the extreme precipitation events are generally

occurring in the wet season, while the small and moderate events primarily occur in the dry seasons. Prominent decreases in Rx5day are observed year-round at high latitudes consistent with general drying under both geoengineering scenarios. Fig. 8g, 8h shows that the zonal means are noisier than for TNn and TXx. The results look much more complex than the temperature extreme indices in Fig. 6h and 7h. The general effect is that the tropical regions (30ºS-30ºN) are more strongly affected by aerosol injection

than by solar dimming. The mid-latitude Rx5day is more effectively changed by stratospheric aerosol injection geoengineering year-round, especially in the Northern Hemisphere. Except for summertime polar areas, solar dimming geoengineering is relatively more effective year-round at high-latitudes, especially in the Southern Hemisphere. Ferraro et al. (2014) found that the tropical overturning circulation weakens in response to geoengineering with stratospheric sulphate aerosol injection due to

radiative heating from the aerosol layer, but geoengineering simulated as a simple reduction in total solar irradiance do not capture this effect. Therefore, a relatively large tropical precipitation perturbation occurs under stratospheric aerosol injection. On the other hand, the meridional distribution of the sulphate aerosols is handled different between the models (as outlined in Section 3.1), which also contributes the noisier Rx5Day pattern showing in Fig. 8d, 8g and 8i. Four of the six models (BNU-ESM, CanESM2,

MIROC-ESM and NorESM1-M) analysed in our study use the AOD prescribed to mimic the one-fourth of the 1991 eruption of Mount Pinatubo, but with different AOD meridional distribution, particle effective radii, and standard deviations of their log-normal size distribution (Kashimura et al., 2017). Another two models (GISS-E2-R and HadGEM2-ES) adopt different stratospheric aerosol schemes to simulate the sulphate AOD.

Stratospheric sulphate aerosols result in heating of the stratosphere, particularly in the tropics, (e.g., Tilmes et al., 2009). Changes in heating rates in the stratosphere and at the tropopause would directly change the tropospheric lapse rate, likely altering the stability of the atmosphere, relative humidity and hence the hydrological cycle. The Northern Hemisphere peak tropical cyclone (TC) season is August through October, and the Southern Hemisphere season is January through March. Interestingly, the Southern Hemisphere ocean basins (5-20°S) where TCs are generated are red in Fig. 8i during the TC season, while in the Northern Hemisphere the TC basins are blue in their TC season. This suggests a dichotomy between the hemispheres insofar as the type of geoengineering that may moderate tropical storms and hurricanes: these are more effectively moderated in the Northern Hemisphere by G4 stratospheric aerosol injection and in the Southern Hemisphere by G1 solar dimming. We have no mechanism for this response, but we note that the response of TC varies between basins with notable hemispheric differences in response to G4 and rcp45 (Wang et al., 2018).

Analysis of the 1991 Pinatubo and 1982 El Chichón volcanic eruptions by Evan (2012) revealed significant reduction in TC number ($p<0.01$) in Atlantic hurricane frequency duration and intensity in the three following seasons compared with the three prior to the eruptions. This corresponds with reduced cyclogenesis in the region 8°-20°N during July-November, driven by decreases in sea surface temperatures of about 0.8°C and stratospheric warming (at 70 hPa) of about 3°C caused by the volcanic aerosol direct effect. The G4 experiment is equivalent to about one-quarter of the 1991 Pinatubo eruption, so the effects would be much weaker, consistent with the modest changes seen in Fig. 8f. The greater effectiveness of G4 stratospheric aerosol than G1 solar dimming in changing Rx5day (Fig. 8i) during July-November in the northern tropics is suggestive that both the sea surface temperature reduction and the stratospheric heating are playing significant roles in changing tropical cyclogenesis.

In summary, the normalized zonal mean annual responses in TNn, TXx and to a lesser extent for Rx5day show similar meridional structure and magnitude for solar dimming and stratospheric aerosol injection geoengineering (Fig. 6h, 7h, 8h), with the exception of Northern Hemisphere high latitudes where the two geoengineering methods show different effectiveness in moderating the seasonality of TNn and TXx.

**4 Summary and conclusions**

We have compared the impacts of reduced solar radiation (G1) and stratospheric aerosol injection (G4) on temperature and precipitation extremes in corresponding reference experiments (abrupt4×CO2 and rcp45, respectively), particularly their spatial and temporal patterns. Most previous studies comparing

solar dimming and stratospheric aerosol SRM have concentrated on the climate mean response (Jones et al., 2011; Niemeier et al., 2013; Ferraro et al., 2014). Curry et al. (2014) examined the effect of G1 geoengineering on the same metrics of extreme temperature and precipitation response (both magnitude and duration) as examined here, but did not compare the responses of solar dimming and stratospheric aerosol injection that we focus on in this paper.

Despite large difference in the magnitude of the response induced by the two geoengineering schemes (which is somewhat larger than the ratio of the input forcings), our results show that the patterns of extreme high and low temperature in solar dimming and stratospheric aerosol injection geoengineering schemes are geographically similar, with regional differences mostly over high latitudes. Solar dimming SRM is relatively more effective in reducing night-time temperatures (TNn) in high-latitude summer,

especially in the Arctic. There are much smaller differences in the effectiveness of aerosol and dimming SRM for the warmest day (TXx), though high latitude winters are more affected by stratospheric aerosols than solar dimming.

As reflected by the wettest consecutive five days index Rx5day, both SRM methods have a moderating effect on extreme precipitation during the hurricane/typhoon seasons for both hemispheres. Stratospheric

aerosol injection is more effective at reducing precipitation during the Northern Hemisphere TC season, while months outside the hurricane season are wetter under solar dimming, and vice versa in the Southern Hemisphere. Despite their different responses, both G1 and G4 moderate Rx5day in the cyclone season while increasing it other months, thus both schemes affect tropical cyclogenesis. Relative differences under both SRM methods are larger in precipitation extremes than for temperature extremes. This may

be because, in addition to the cooling of sea surface temperatures facilitated by both solar dimming and stratospheric aerosol injection, stratospheric aerosol injection heats the stratosphere via absorbing near infrared and longer wavelengths radiation (Lohmann and Feichter, 2005). This mechanism is present in all models analyzed here. This finding suggests that models that rely only on parameterizing hurricane numbers and intensity by surface temperatures (Moore et al., 2015) are likely to underestimate the impact

of stratospheric aerosol geoengineering compared with comparable amounts of solar dimming, though there are very large differences in how both greenhouse gas warming and stratospheric aerosol injection affects cyclogenesis across the different tropical basins (Wang et al., 2018).

Davis et al (2016) and Smyth (2017) examined the changes in the mean state of the tropical Hadley cells to GHG forcing and the G1 scenario. They note that the poleward expansion of the Hadley cell occurs under the GHG forcing, but under G1 it is indistinguishable from the preindustrial control state, and moreover find that the ITCZ is reduced in its seasonal migration amplitude under G1 but not GHG forcing. Further analysis of the Hadley and Walker cell intensities under G1 (Guo et al., 2018) shows that the Hadley circulation is reduced under G1 relative to piControl, but that changes under GHG forcing are rather more complex, affecting also the higher latitude Ferrel cells. Thus, some of the relative differences seen in the extreme indices around the tropics may reflect a tendency of geoengineering to mitigate changes in the Hadley cell caused by GHG forcing.

The hydrological cycle strength weakens under both types geoengineering. In our analysis, the global mean precipitation decrease per Kelvin is stronger in response to G4 stratospheric aerosol than to G1 solar dimming. This is consistent with a previous study by Niemeier et al. (2013), in which impacts on energy balance and hydrological cycle by three different solar geoengineering schemes are examined. The differences between stratospheric aerosol injection and solar dimming are influenced strongly by the absorption of longwave radiation by aerosols, this atmospheric heating imbalance could further stabilize the troposphere and lead to stronger precipitation reduction under stratospheric aerosol injection than under solar dimming (Niemeier et al., 2013). Recently Xia et al. (2018) found precipitation and evaporation changes are very similar under sulphate and solar dimming geoengineering schemes using the full tropospheric and stratospheric chemistry version of the Community Earth System Model (CESM). This is different from previous studies by Niemeier et al. (2013) and Ferraro et al. (2014), who found that the sulphate geoengineering has larger effect on the hydrological cycle. Xia et al. (2018) suggested that the column ozone change could possibly play an important role in a fully coupled atmosphere–chemistry model by changing radiative forcing and atmospheric lapse rate, while in Niemeier et al. (2013) and Ferraro et al. (2014) the same prescribed ozone was used in all scenarios. In our study, according to the scaling of global mean precipitation reduction to mean temperature change, all models show a relatively larger reduction of global mean precipitation per Kelvin under stratospheric aerosol injection than under solar dimming, which is consistent with Niemeier et al. (2013) and Ferraro

et al. (2014). Among six GeoMIP models used here, only GISS-E2-R model calculates ozone for its G4 simulation, other models and experiments all use prescribed ozone. Therefore, we cannot diagnose ozone's roles as suggested by Xia et al. (2018).

Compared with solar dimming SRM, aerosol SRM has larger differences between models and a much lower signal-to-noise ratio, although the aerosol geoengineering applied was of a much smaller magnitude than the solar dimming. Aerosol SRM was relatively less effective in increasing cold spell duration and decreasing warm spell duration in equatorial oceans than solar dimming, consistent with a relatively smaller cooling effect in coldest day and warmest night in equatorial oceans than in adjacent regions. The reduced cooling effect in equatorial oceans in aerosol SRM may result from the smaller reduction in shortwave radiation flux at the top of atmosphere in aerosol SRM in these regions.

Climate extremes are more readily perceived by society and can have more immediate economic and social impacts than changes in mean climate (IPCC, 2012). Yet, the ETCCDI extreme climate indices may not reflect what are considered "extreme events" by the public. These would include events such as typhoons, severe heatwaves etc., that may occur much less frequently, but are of higher intensity, than the thresholds represented by the indices used here. The downscaling and impact modelling required to assess geoengineered climate effects has so far been limited to a study of Atlantic hurricane storm surge size and frequency (Moore et al., 2015), but such studies are a clear focus of ongoing research. More climate models with various aerosol parameterization schemes are certainly needed to describe the extreme tails of simulated climate variables. These extremes are incompletely sampled from 40-year long periods of model runs, but may be explored more thoroughly by specific impact models driven by the thermodynamic state of the climate system (Emanuel, 2013), and by planned extensions to the G1 experiment outlined under GeoMIP6 (Kravitz et al., 2015).

**Acknowledgements**

We thank all participants of the Geoengineering Model Intercomparison Project and their model development teams, the CLIVAR/WCRP Working Group on Coupled Modelling for endorsing GeoMIP, and the scientists managing the Earth System Grid data nodes who have assisted with making GeoMIP output available. Research was funded by the National Basic Research Program of China grant number 2015CB953600. HK and SW were supported by the SOUSEI Program, MEXT, Japan. CC is supported

by the NSERC-funded Canadian Sea Ice and Snow Evolution Network. HM was supported by Research Council of Norway grant 229760/E10, and Sigma2 HPC resources hexagon and norstore (accounts nn9812k, nn9448k, NS9033K). MIROC-ESM simulations were conducted using the Earth Simulator. We thank Andy Jones for model development and comments on the manuscript. The Pacific Northwest National Laboratory is operated for the U.S. Department of Energy by Battelle Memorial Institute under contract DE-AC05-76RL01830.

**Appendix:** TOA net radiation flux differences of G1-abrupt4×CO2 and G4-rcp45 for each model

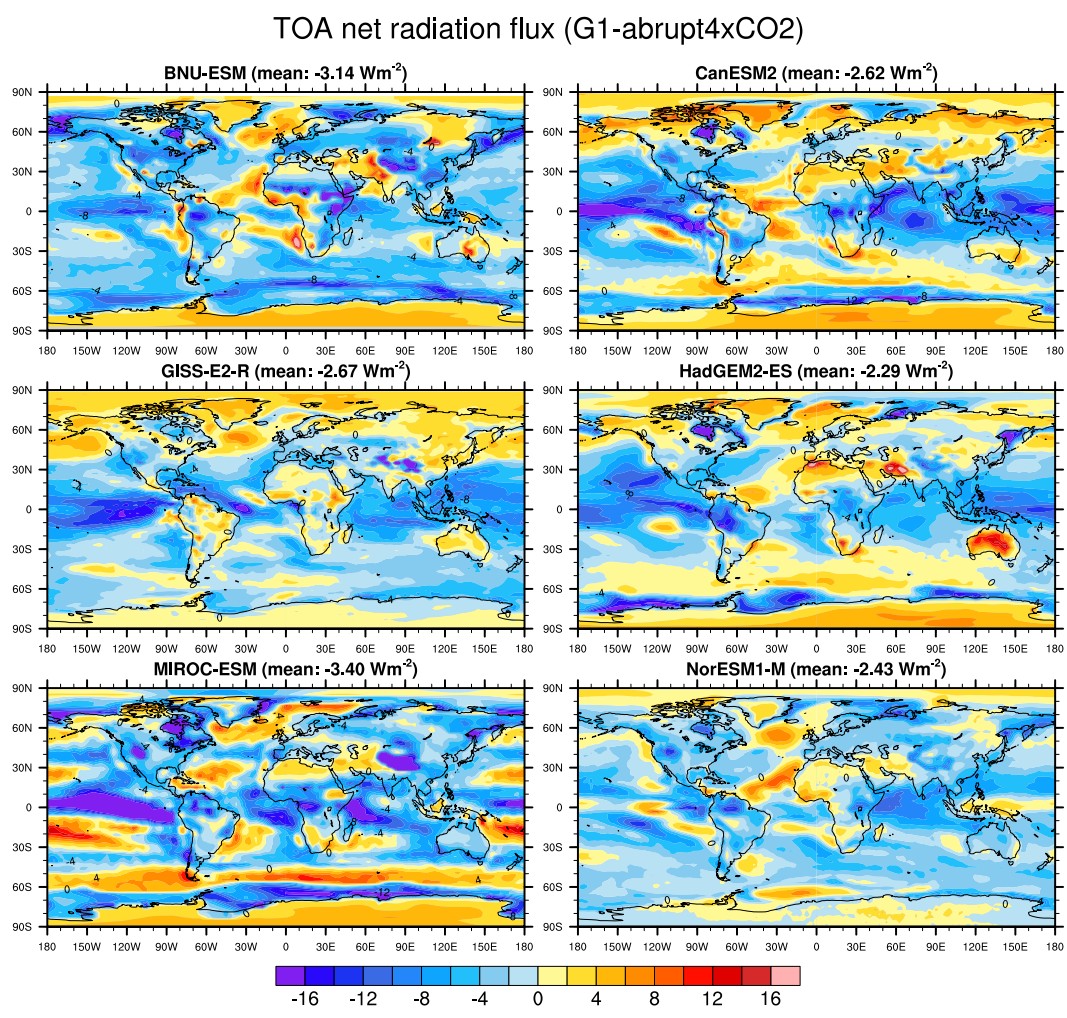

**Figure A1: Geographical distributions over the 40-year analysis periods of the differences G1 - abrupt4×CO2 for TOA net radiation flux for BNU-ESM, CanESM2, GISS-E2-R, HadGEM2-ES, MIROC-ESM and NorESM1-M.**

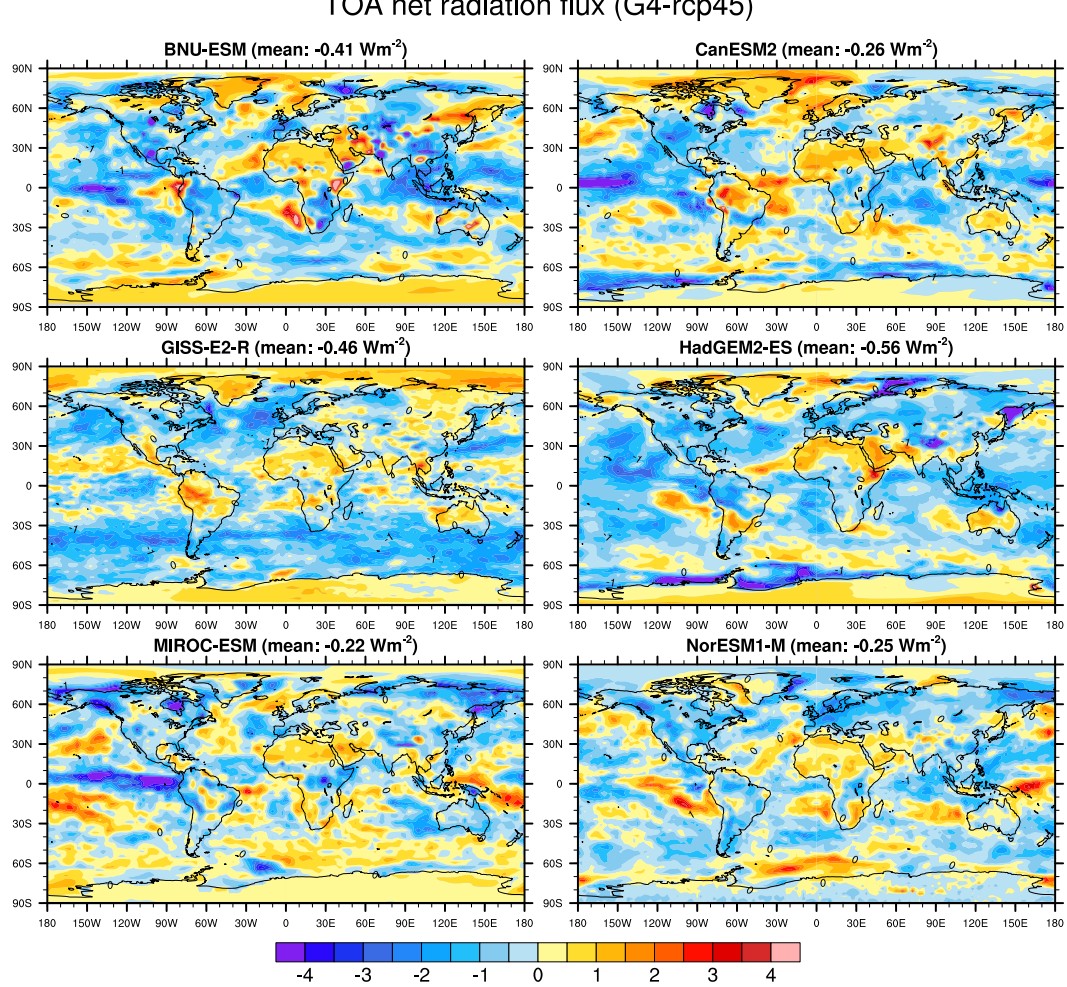

**Figure A2: Geographical distributions over the 40-year analysis periods of the differences G4 – rcp45 for TOA net radiation flux for BNU-ESM, CanESM2, GISS-E2-R, HadGEM2-ES, MIROC-ESM and NorESM1-M.**

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

**Table 1: GeoMIP Models used in this study**

| No. | Model | Institution | Resolution (Lon×Lat Level) |
|---|---|---|---|
| 1 | BNU−ESM (Ji et al. 2014) | Beijing Normal University, China | 2.8°×2.8° L26 |
| 2 | CanESM2 (Arora et al., 2011) | Canadian Centre for Climate Modelling, Canada | 2.8°×2.8° L35 |
| 3 | GISS-E2-R (Schmidt et al. 2011) | Goddard Institute for Space Studies, USA | 2.5°×2.0° L40 |
| 4 | HadGEM2−ES (Collins et al. 2011) | Met Office Hadley Centre, UK | 1.875°×1.25° L40 |
| 5 | MIROC−ESM (Watanabe et al. 2011) | AORI, NIES, JAMSTEC, Japan | 2.8°×2.8° L80 |
| 6 | NorESM1-M (Bentsen et al. 2013) | University of Oslo, Norway | 1.9°x2.5° L26 |

**Table 2: Indices of climate extremes**

| Index | Description | Definition | Units |
|---|---|---|---|
| TNn | Coldest daily Tmin | Annual minimum value of daily minimum temperature | ℃ |
| TXx | Warmest daily Tmax | Annual maximum value of daily maximum temperature | ℃ |
| Rx5day | Wettest consecutive five days | Maximum of consecutive 5-day (cumulative) precipitation amount | mm |
| CSDI | Cold spell duration | Number of consecutive days (> 6 days) when daily minimum temperature falls below the 10th percentile of piControl | days |
| WSDI | Warm spell duration | Number of consecutive days (> 6 days) when daily maximum temperature falls above the 90th percentile of piControl | days |
| CDD | Consecutive dry days | Maximum number of consecutive days when precipitation < 1 mm | days |

**Table 3: Differences and ratios in means and climate extreme indices over the 40-year analysis period.**

| Experiments | Indices | Land | Ocean | Global |
|---|---|---|---|---|
| G1 − abrupt4 × CO2 | TNn(°C) | -6.4 | -4.5 | -5.1 |
| | TXx(°C) | -6.2 | -3.7 | -4.4 |
| | Rx5day(mm) | -12.3 | -13.4 | -13.1 |
| | TOA net radiation flux(Wm$^{-2}$) | -1.6 | -3.2 | -2.8 |
| | Mean T(°C) | -5.6 | -3.7 | -4.3 |
| | Mean P(mm a$^{-1}$) | -81.0 | -106.4 | -98.9 |
| G4 − rcp45 | TNn(°C) | -0.9 | -0.6 | -0.7 |
| | TXx(°C) | -0.8 | -0.5 | -0.6 |
| | Rx5day(mm) | -1.6 | -1.9 | -1.8 |
| | TOA net radiation flux(Wm$^{-2}$) | -0.2 | -0.4 | -0.4 |
| | Mean T(°C) | -0.7 | -0.5 | -0.5 |
| | Mean P(mm a$^{-1}$) | -8.2 | -16.6 | -14.1 |
| $\dfrac{G1 - abrupt4 \times CO2}{G4 - rcp45}$ | TNn(°C) | 6.9 | 7.6 | 7.3 |
| | TXx(°C) | 7.5 | 7.7 | 7.6 |
| | Rx5day(mm) | 7.8 | 7.2 | 7.3 |
| | TOA net radiation flux(Wm$^{-2}$) | 8.3 | 7.6 | 7.7 |
| | Mean T(°C) | 7.9 | 8.3 | 8.1 |
| | Mean P(mm a$^{-1}$) | 9.8 | 6.4 | 7.0 |