# Peer review of "Extreme temperature and precipitation response to solar dimming and stratospheric aerosol geoengineering"

_Atmospheric Chemistry and Physics, 2018_

## Referee Comment (RC1) · Anonymous Referee #1 · 20 Apr 2018

Review on:

**Extreme temperature and precipitation response to solar dimming and stratospheric aerosol geoengineering**

Duoying Ji et al

April 20, 2018

The authors analyze GeoMIP experiments G1 and G4 on extreme values. So far, this has only be done for G1 results (solar dimming) but not for injection of sulfate into the stratosphere. They try to estimate differences in the efficiency of the two models, despite the large differences in the forcing.

The paper is mostly well written. I recommend publication when the following after the authors addressed the following comments and questions.

**General:**

Some results in Figures 6, 7 and 8 are not clearly described and need further explanations. Also the amount of single figures within these figures can be reduced.

The paper will gain if the differences in the forcing of the two experiments are clearly stated in the beginning. Mean values of temperature and precipitation might also be helpful for the reader.

The hypothesis given in the abstract have to be better based on results and further explanations. You should avoid a propbably in the abstract. This is the place for new results. So yave should base the probably on results and you can give a clear answer. E.g. show that you see the claimed respose of the stratospheric dynamic in the results, like a change in the polar vortex.

The forcings of the two experiment are very different. Therefore I recommend to normalize the results when possible add add this to the figures.

The analyses in (Xia et al., 2017) and Niemeier et al (2013) on the differences of solar and aerosol forcing may provided some answers to these results, especially on the hyrological

cycle.

**Specific comments**

Line 1: A hypothesis like this is not great in the abstract. You may better provide results.

Line 7: Again, better avoid probably in the abstract. You can name the differences here in case they are significant.

Line 24: In case you want to cite cloud brightening here, you should add Alterskjer et al (2013) with G5. But cloud brightning is not relevant for your topic.

Line 5: Neither Geoengineering nor the future produce warming or cooling. Please rephrase.

Line 9: A number of the mean forcing would be good.

Line 21: It is not clear to me if you calculate the PDF for the ensemble mean or for each model and average thereafter.

Line 24/25: Reformulate to include both experiments.

1st paragraph: Forcing differences to the control run are much stronger in G1 than in G4. Readers not familiar with this experiments may not realize this right away. You may add a figure to highlight this differences, e.g. time series or give a number of the forcings earlier in the text. OK, you do later. Maybe you reorder your paragraphs.

How different are the PDF results to Curry et al?

Line 14: TOA forcing (SW +LW) is more general than SW at the surface. For sulfate CE the LW part is important. In case TOA forcing adds some information, you may add this to Figure 2.

Line 25: In case you talk about single models you may take into account that the forcing from the sulfate layer may be very different between the models (see (Pitari et al., 2013))

Line 2/3: Climate under G1 is pre-industrial while G4 is 2020. This can play a role here.

Figure 2: I wonder about the hatching in c). Only small areas with values close o one are significant. Is this correct and do you have an explanation?

Figure 3: You may add some words that Rx5days and G4 is partly positive, increase in precipitation, which indicates nicely the climate variability.

Line 2: Do mean values of p and t scale the same in G1 and G4?

Fig 4: Please add normalized values here. You may normalize with mean T and P or with the TOA forcing difference G1-4xCO2 and G4-RCP45.

I am not sure I understand what you did in Fig 4i) You may add an equation. Again, hatching and significance seem not to fit. Are this area not significant? Otherwise your area with significant results is very small and you should discuss the relevance of your results in this case.

Line 6 to 16: You give a list of related topics here but I miss the clear relation to your results.

Line 17 to 24: The paper would gain of you can explain this feature better

Line 28: 'Except Eastern China' this is a very small area. I guess this is a regional feature which is usually not very well represented in the models. Otherwise an explanation would be great.

Line 29: Better use solar dimming and stratospheric aerosol here also, so decide for one naming and stick to it.

Line 11 to 13: I got lost. Can I see this in any figure?

Line 17/18: As you mention them, you may summarize the more complex interactions.

Line 28: Can you show this in a figure, link to existing figure or literature. In general, the small signal to noise ration is important for G4.

Line 1 to 5: The forcing varies stronger in G4 between the models than in G1. In case a model simulates of provides most sulfate in the tropics cooling at the poles would be relatively low. You may show TOA of the single models in an appendix.

Line 9: Better cite Schmidt et al, 2012 or Kravitz et al, 2013 here. They have shown the shift of the ITCZ.

Line 15:Any explanation for the different results over central Asia. This results is also different to Aswathy et al (2015).

Fig 6: I am not sure I understood how you normalize the right column. I understood (i) as b/e and (j) as c-f. But than the pattern should be similar.

Increase the contrast of Fig (c) and (f), thus darker blue.

Can you add significance to (i) and (j)? Values around 1 and 0 should be white (also in Fig 7 and 8)

Line 8: Which figure? Cannot see this in 6c) or 6f).

Line 18: 'affected more by solar dimming....' Where? It is largely blue. I may not get all positive and negative changes right. You may explain this a bit more.

Line 19: Quit dark red around 60S

Line 9 to 15: You are on quite weak ground here. The polar stratospheric vortex may weaken due to less sea ice and more wave propagation, sulfate warming the stratosphere may strengthen the vortex (Ferraro et al, 2014) etc.

Line 16: Yes, but this depends also on the meridional distribution of the aerosols which is not given in the paper and may be different between the models.

Line 21: 'at high latitudes winter solar dimming ...' How can this be? There is no sunlight, so no SW reduction. < Line 23/24: 'Stratospheric vortex' I doubt that you see this with significant signals in the models even it seems to be a reasonable explanation. E.g. Driscoll et al. (2012) showed that climate model do not represent stratospheric dynamic repose of volcanic eruptions very well. You may show the zonal wind (DJF) and significance of the single models to approve the hypothesis.

Line 1 to 6: Does this result it to Jones et al. (2017)?

References

Some are missing in the list. Please check.

**References**

Driscoll, S., Bozzo, A., Gray, L. J., Robock, A., and Stenchikov, G.: Coupled Model Intercomparison Project 5 (CMIP5) simulations of climate following volcanic eruptions, Journal of Geophysical Research: Atmospheres, 117, doi:10.1029/2012JD017607, 2012.

Jones, A. C. J. M. H., Dunstone, N., Hawcroft, K. E. M. K., and Jones, K. H. A.: Impacts of hemispheric solar geoengineering on tropical cyclone frequency, Nature Communications, 8, 1383, doi:10.1038/s41467-017-01606-0, 2017.

Pitari, G., Aquila, V., Kravitz, B., Robock, A., Watanabe, S., Luca, N. D., Genova, G. D., Mancini, E., Tilmes, S., and Cionni, I.: Stratospheric ozone response to sulfate geoengineering: Results from the Geoengineering Model Intercomparison Project (GeoMIP), Journal of Geophysical Research, 119, 2629–2653, doi:10.1002/2013JD020566, 2013.

Xia, L., Nowack, P. J., Tilmes, S., and Robock, A.: Impacts of stratospheric sulfate geoengineering on tropospheric ozone, Atmospheric Chemistry and Physics, 17, 11 913–11 928, doi:10.5194/acp-17-11913-2017, URL `https://www.atmos-chem-phys.net/17/11913/2017/`, 2017.

---

## Referee Comment (RC2) · Anonymous Referee #2 · 22 Apr 2018

General Comments: In this manuscript, the authors analyzed the extreme values of climate indicators under 2 different solar radiation management scenarios G1 and G4. They took extreme index by ETCCDI and applied it on temperature and precipitation. The authors tried to find the differences and similarities on the global impact of two SRM experiment. And also tried to analysis the differences among the model.

This manuscript is novel and further complete the understanding of SRM. The structure is also well organized. I recommend the manuscript for publication though some of the comments still should be fixed or rephrased.

Specific Comments: 1. The significant regions in Fig. 2c and 3c,f,j need further de-

scriptions on calculation process; 2. The uncertainty reason present on abstract may not be proper here. Rephrase the word may be better. 3. On Page 15, Line 1-6, this paragraph are not linked so well with the context. There are also no further analysis on the daily rain types. Further explanation and graphs would be better.

Minor comments: 1. P2 L18: Missed ref. Lathan et al.

2. P3 L12: Missed ref. Niemeier et al.

3. P5 L1: The estimate of CSDI and WSDI is applied on ensemble mean temperature or mean CSDI/WSDI?

4. P8 L24-27: It is not clear for me about the relations between different models and the geoengineering impact. Further expression would be better.

5. P12 L2: May be I got missed but I'm not sure what the 'case' indicate.

6. P13 L28: Eastern China in Fig4 seems no special around the globe, this part may need further explanation.

7. P14 L10-13: The reduction of Rx5day is whether a result from Curry et al., 2014 or from the paper result? Further explanation would be better.

8. P15 L1-6: The paragraph may not fully link with the context and there is no graphs or tables to support the statistics.

---

## Author Comment (AC1) · 22 Jun 2018

**Response to Review of "Extreme temperature and precipitation response to solar dimming and stratospheric aerosol geoengineering" by D. Ji et al.**

We first thank the referee for his/her insightful comments, which helped us clarify and greatly improve the paper. In the reply, the referee's comments are in *italics*, our response is in normal and changes to the text are shown in blue.

***Anonymous Referee #1***

*The authors analyze GeoMIP experiments G1 and G4 on extreme values. So far, this has only be done for G1 results (solar dimming) but not for injection of sulfate into the stratosphere. They try to estimate differences in the efficiency of the two models, despite the large differences in the forcing.*
*The paper is mostly well written. I recommend publication when the following after the authors addressed the following comments and questions.*

***General:***
*Some results in Figures 6, 7 and 8 are not clearly described and need further explanations. Also the amount of single figures within these figures can be reduced.*

Reply: Thanks. We revise Figures 6, 7 and 8 and remove 3 single figures in total, we also revise the relevant text to comply with these new figures. In new Figures 6h, 7h and 8h, we show the annual zonal mean differences between normalized solar dimming and stratospheric aerosol geoengineering effects instead of the ratios between them. Most of the ratios showed in previous Figure 6h, 7h and 8h are close to value of one, but very large values occur when the denominators close to zero. With changing to show the differences between two normalized effects of geoengineering, we can avoid of this situation and better present the differences between solar dimming and stratospheric aerosol. Similar to new Figure 6h, 7h and 8h, we show the monthly climatological differences of normalized solar dimming effects and stratospheric aerosol effects in new Figure 6i, 7i and 8i. The previous Fig. 6j, 7j and 8j are removed in new figures as they deliver similar messages as Fig. 6i, 7i and 8i.

We define all normalization methods in Section 2: Data and methods, see the reply to another of the "general comment" below.

The following are new Figures 6, 7 and 8:

[Figure]

**Figure 6: Absolute difference of annual zonal mean in the extreme low temperature TNn (left column), Normalized difference with respect to annual zonal mean (middle column) and monthly zonal mean (right column) in TNn: (a), (b), (c) G1 – abrupt4×CO2, (d), (e), (f) G4 – rcp45, (g) the ratio between absolute G1 – abrupt4×CO2 and G4 – rcp45, (h) the annual zonal mean difference between normalized G1 – abrupt4×CO2 and G4 – rcp45, (i) the monthly zonal mean difference between normalized G1 – abrupt4×CO2 and G4 – rcp45 taken over the 40-year analysis period. In panel (i) red colours indicate relatively greater changes with G4 and blue colours with G1, stippling indicates regions where fewer than 5 of 6 models agree on the sign of the model response. 3×3-point smoothing was applied to the seasonal-latitude change.**

[Figure]

**Figure 7: Absolute difference of annual zonal mean in the extreme high temperature TXx (left column), Normalized difference with respect to annual zonal mean (middle column) and monthly zonal mean (right column) in TXx: (a), (b), (c) G1 – abrupt4×CO2, (d), (e), (f) G4 – rcp45, (g) the ratio between absolute G1 – abrupt4×CO2 and G4 – rcp45, (h) the annual zonal mean difference between normalized G1 – abrupt4×CO2 and G4 – rcp45, (i) the monthly zonal mean difference between normalized G1 – abrupt4×CO2 and G4 – rcp45 taken over the 40-year analysis period. In panel (i) red colours indicate relatively greater changes with G4 and blue colours with G1, stippling indicates regions where fewer than 5 of 6 models agree on the sign of the model response. 3×3-point smoothing was applied to the seasonal-latitude change.**

[Figure]

**Figure 8: Absolute difference of annual zonal mean in the extreme precipitation Rx5day (left column), Normalized difference with respect to annual zonal mean (middle column) and monthly zonal mean (right column) in Rx5day: (a), (b), (c) G1 – abrupt4×CO2, (d), (e), (f) G4 – rcp45, (g) the ratio between absolute G1 – abrupt4×CO2 and G4 – rcp45, (h) the annual zonal mean difference between normalized G1 – abrupt4×CO2 and G4 – rcp45, (i) the monthly zonal mean difference between normalized G1 – abrupt4×CO2 and G4 – rcp45 taken over the 40-year analysis period. In panel (i) blue colours indicate relatively greater changes with G4 and red colours with G1, stippling indicates regions where fewer than 5 of 6 models agree on the sign of the model response. 3×3-point smoothing was applied to the seasonal-latitude change.**

*The paper will gain if the differences in the forcing of the two experiments are clearly stated in the beginning. Mean values of temperature and precipitation might also be helpful for the reader.*

Reply: Thanks for this constructive suggestion. We reorder the results of "TOA net radiation flux" and "Probability distribution of monthly temperature and precipitation" with introducing the radiative forcing results firstly. For the differences in the forcing of the two experiments, we add the following part in the "Introduction":

Both methods would cool Earth's surface by reducing sunlight reaching the surface, either by aerosols reflecting sunlight or by artificially reducing the solar constant in climate models. The injected stratospheric aerosols under G4 not only scatter shortwave radiation, also absorb near infrared and longer wavelengths radiation (Lohmann and Feichter, 2005). The differences between stratospheric aerosol injection and solar dimming are influenced strongly by the absorption of longwave radiation by aerosols, this atmospheric heating imbalance could further stabilize the troposphere and lead to stronger precipitation reduction under stratospheric aerosol injection than under solar dimming (Niemeier et al., 2013).

Lohmann, U., and Feichter, J.: Global indirect aerosol effects: A review, Atmos. Chem. Phys., 5, 715–737, doi:10.5194/acp-5-715-2005, 2005.

Niemeier, U., Schmidt, H., Alterskjær, K., and Kristjánsson, J. E.: Solar irradiance reduction via climate engineering: Impact of different techniques on the energy balance and the hydrological cycle, J. Geophys. Res. Atmos., 118, 11905–11917, doi:10.1002/2013JD020445, 2013.

In the "TOA net radiation flux" part, we add:

The forcing of the G1 solar dimming and G4 stratospheric aerosol injection experiments are quite different, there can be a difference in the mean and extreme climate responses. The multi-model ensemble mean net radiation flux at top of atmosphere (TOA) is 2.76Wm$^{-2}$ and 0.004 Wm$^{-2}$ for the abrupt×4CO2 and G1 experiments, and 1.63 Wm$^{-2}$ and 1.27 Wm$^{-2}$ for the rcp45 and G4 experiments during their 40-year analysis period. Therefore, the G1 solar dimming and G4 stratospheric aerosol injection exert a reduction of 2.76 Wm$^{-2}$ and 0.36 Wm$^{-2}$ for net radiation fluxes at TOA respectively. The differences of mean net radiation flux at TOA over land and ocean between two geoengineering experiments and their reference experiments are show in Table 3.

We also revise Table 3 to present changes of net radiation flux at TOA instead of net shortwave radiation flux.

In the "Probability distribution of monthly temperature and precipitation" part we add:

The G1 solar dimming and G4 stratospheric aerosol injection geoengineering greatly affected the mean climate states. The annual mean surface air temperatures are 291.0 K and 286.7 K for abrupt4×CO2 and G1 experiments, 288.8 K and 288.3 K for rcp45 and G4 experiments respectively during their 40-year analysis period. The global hydrological strength is likewise reduced; the annual precipitation totals are 1125.8 mm and 1026.9 mm for abrupt4×CO2 and G1 experiments, 1098.4 mm and 1084.3 mm for rcp45 and G4 experiments (Table 3).

*The hypothesis given in the abstract have to be better based on results and further explanations. You should avoid a propbably in the abstract. This is the place for new results. So yave should base the probably on results and you can give a clear answer. E.g. show that you see the claimed respose of the stratospheric dynamic in the results, like a change in the polar vortex.*

Reply: Thanks for your constructive comment. We revise the abstract as the following:

We examine extreme temperature and precipitation under two potential geoengineering methods forming part of the Geoengineering Model Intercomparison Project (GeoMIP). The solar dimming experiment G1 is designed to completely offset the global mean radiative forcing due to a CO$_2$-quadrupling experiment (abrupt4×CO2), while in GeoMIP experiment G4, the radiative forcing due to the representative concentration pathway 4.5 (RCP4.5) scenario is partly offset by a simulated layer of aerosols in the stratosphere. Both G1 and G4 geoengineering simulations lead to lower minimum temperatures (TNn) at higher latitudes, and on land primarily through feedback effects involving high latitude processes such as snow cover, sea ice and soil moisture. There is larger cooling of TNn and maximum temperatures (TXx) over land compared with oceans, and the land-sea cooling contrast is larger for TXx than TNn. Maximum 5-day precipitation (Rx5day) increases over subtropical oceans,

whereas warm spells decrease markedly in the tropics, and the number of consecutive dry days decreases in most deserts. The precipitation during the tropical cyclone (hurricane) seasons becomes less intense, whilst the remainder of the year becomes wetter. Stratospheric aerosol injection is more effective than solar dimming in moderating extreme precipitation (and flooding). Despite the magnitude of the radiative forcing applied in G1 being ~7.7 times larger than in G4, and differences in the aerosol chemistry and transport schemes amongst the models, the two types of geoengineering show similar spatial patterns in normalized differences of extreme temperatures changes. Large differences mainly occur at northern high latitudes, where stratospheric aerosol injection more effectively reduces TNn and TXx. While the pattern of normalized differences of extreme precipitation is more complex than that of extreme temperatures, generally stratospheric aerosol injection is more effective in reducing tropical Rx5day, while solar dimming is more effective over extra-tropical regions.

*The forcings of the two experiment are very different. Therefore I recommend to normalize the results when possible add add this to the figures.*

Reply: Thanks. In our new Figure 1 (we reorder some paragraph, previous Fig.2 is labelled as Fig. 1 now) and Figure 4, we show the normalized results. We normalize the values of each grid from the differences of G1-abrupt4xCO2 according to the global average of G1-abrupt4xCO2, same for G4-rcp45. With these normalized results, we present the difference between normalized G1-abrupt4xCO2 and G4-rcp45 instead of the ratio between non-normalized G1-abrupt4xCO2 and G4-rcp45 to avoid large unrealistic values. In Figure 6, 7 and 8, we also show the differences of zonally normalized results in several single figures instead of ratios between non-normalized fields. We define all normalization methods in Section 2: Data and methods as the following:

2.3 Normalization methods

There are large differences in forcing between the G1 solar dimming and G4 stratospheric aerosol injection geoengineering schemes. The mean and extreme climates under the two type geoengineering are quite different as will be shown below. To aid the comparisons, we adopt the following normalization methods to compare spatially relative effectivities between solar dimming and stratospheric aerosol injection.

The normalized global spatial effects of solar dimming or stratospheric aerosol injection are defined as the grid mean difference relative to the global mean difference:

$$< X^{geo} - X^{ref} > = \frac{\overline{X}_{grid}^{geo} - \overline{X}_{grid}^{ref}}{|\overline{X}_{global}^{geo} - \overline{X}_{global}^{ref}|}$$

where the operator $<>$ denotes the normalized grid value, X is TXx, TNn, Rx5day or other climate field, an overbar denotes the average of each grid cell or the global average, the absolute operator $||$ in the denominator of the right term preserves the sign of the geoengineering anomaly. The superscript "geo" represents geoengineering experiments of G1 solar dimming or G4 stratospheric aerosol injection, the superscript "ref" represents the reference experiments of abrupt4×CO2 or rcp45.

To normalize zonal mean difference in the climate extreme indices relative to the global mean difference, we use a similar formula:

$$< X^{geo} - X^{ref} > = \frac{\overline{X}_{zonal}^{geo} - \overline{X}_{zonal}^{ref}}{|\overline{X}_{global}^{geo} - \overline{X}_{global}^{ref}|}$$

where the operator $\diamond$ denotes the normalized zonal mean, an overbar denotes the zonal or global average, the absolute operator || in the denominator of the right term preserves the sign of the geoengineering anomaly.

*The analyses in (Xia et al., 2017) and Niemeier et al (2013) on the differences of solar and aerosol forcing may provided som answers to these results, especially on the hyrological cycle.*

Reply: Thanks for your constructive suggestions. We cite Niemeier et al. (2013) to explain the relatively stronger precipitation reduction under G4 stratospheric aerosol injection than that under G1 solar dimming.

The hydrological cycle strength weakens under both types geoengineering. In our analysis, the global mean precipitation decrease per Kelvin is stronger in response to G4 stratospheric aerosol than to G1 solar dimming. This is consistent with a previous study by Niemeier et al. (2013), in which impacts on energy balance and hydrological cycle by three different solar geoengineering schemes are examined. The differences between stratospheric aerosol injection and solar dimming are influenced strongly by the absorption of longwave radiation by aerosols, this atmospheric heating imbalance could further stabilize the troposphere and lead to stronger precipitation reduction under stratospheric aerosol injection than under solar dimming (Niemeier et al., 2013).

Based on the analysis on scaling of global mean precipitation reduction to mean temperature change (please check our reply to one of your "specific comments"), we add the following part in the "Discussion":

Recently Xia et al. (2018) found precipitation and evaporation changes are very similar under sulfate and solar dimming geoengineering schemes using the full tropospheric and stratospheric chemistry version of the Community Earth System Model (CESM). This is different from previous studies by Niemeier et al. (2013) and Ferraro et al. (2014), who found that the sulfate geoengineering has larger effect on the hydrological cycle. Xia et al. (2018) suggested that the column ozone change could possibly play an important role in a fully coupled atmosphere–chemistry model by changing radiative forcing and atmospheric lapse rate, while in Niemeier et al. (2013) and Ferraro et al. (2014) the same prescribed ozone was used in all scenarios. In our study, according to the scaling of global mean precipitation reduction to mean temperature change, all models show a relatively larger reduction of global mean precipitation per Kelvin under stratospheric aerosol injection than under solar dimming, which is consistent with Niemeier et al. (2013) and Ferraro et al. (2014). Among six GeoMIP models used here, only GISS-E2-R model calculates ozone for its G4 simulation, other models and experiments all use prescribed ozone. Therefore, we cannot diagnose ozone's roles as suggested by Xia et al. (2018).

Xia, L., Nowack, P. J., Tilmes, S., and Robock, A.: Impacts of stratospheric sulfate geoengineering on tropospheric ozone, Atmos. Chem. Phys., 17, 11913-11928, doi:10.5194/acp-17-11913-2017, 2017.

*Specific comments*
*Page 2*
*Line 1: A hypothesis like this is not great in the abstract. You may better provide results.*

Reply: Thanks. Done. Please refer to our previous reply to your general comment.

*Line 7: Again, better avoid probably in the abstract. You can name the differences here in case they are significant.*

Reply: Thanks. Done. Please refer to our previous reply to your general comment.

*Line 24: In case you want to cite cloud brightening here, you should add Alterskjer et al (2013) with G5. But cloud brightning is not relevant for your topic.*

Reply: Thanks. We rephrase this sentence to the following:
Kravitz et al. (2011, 2013c) defined a set of numerical SRM experiments under the Geoengineering Model Intercomparison Project (GeoMIP), comprising solar dimming experiments (G1 and G2), stratospheric aerosol injection simulations (G3 and G4) and marine cloud brightening experiments (G4cdnc, G4sea-salt).

*Page 3*
*Line 5: Neither Geoengineering nor the future produce warming or cooling. Please rephrase.*

Reply: We rephrase this sentence to the following:
Dagon and Schrag (2017) showed that solar geoengineering mitigates extreme heat events from greenhouse warming, though the regional response is variable in part due to varying soil moisture content: soils dry out over the course of the summer as daily maximum temperature increases, and this relationship is strengthened under solar geoengineering.

*Page 4*
*Line 9: A number of the mean forcing would be good.*

Reply: Thanks. We add the following sentence after Line 9:
The global temporally averaged forcing of the G1 solar dimming experiment ranges from -9.6 to -6.4 $Wm^{-2}$, and the G4 stratospheric aerosol experiment ranges from $-3.6$ to $-1.6\,Wm^{-2}$, depending on the model (Schmidt et al., 2012; Kashimura et al., 2017).

Schmidt, H., Alterskjær, K., Bou Karam, D., Boucher, O., Jones, A., Kristjánsson, J. E., Niemeier, U., Schulz, M., Aaheim, A., Benduhn, F., Lawrence, M., and Timmreck, C.: Solar irradiance reduction to counteract radiative forcing from a quadrupling of CO2: climate responses simulated by four earth system models, Earth Syst. Dynam., 3, 63–78, doi:10.5194/esd-3-63-2012, 2012.
Kashimura, H., Abe, M., Watanabe, S., Sekiya, T., Ji, D., Moore, J. C., Cole, J. N. S., and Kravitz, B.: Shortwave radiative forcing, rapid adjustment, and feedback to the surface by sulfate geoengineering: analysis of the Geoengineering Model Intercomparison Project G4 scenario, Atmos. Chem. Phys., 17, 3339–3356, doi:10.5194/acp-17-3339-2017, 2017.

*Page 5*

*Line 21: It is not clear to me if you calculate the PDF for the ensemble mean or for each model and average thereafter.*

Reply: We calculate the PDF for each model and average thereafter. We rephrase the related sentence as the following to make it clear:

We computed the probability density functions (PDFs) of temperature and precipitation for each model and average all models thereafter to get a general idea of the changes in the two geoengineering experiments (G1 and G4) compared to their baseline experiments (abrupt4×CO2 and rcp45).

*Line 24/25: Reformulate to include both experiments.*

Reply: Done.

*Page 8*
*1st paragraph: Forcing differences to the control run are much stronger in G1 than in G4. Readers not familiar with this experiments may not realize this right away. You may add a figure to highlight this differences, e.g. time series or give a number of the forcings earlier in the text. OK, you do later. Maybe you reorder your paragraphs.*
*How different are the PDF results to Curry et al?*

Reply: Thanks for your constructive suggestion. We reorder the results of 3.1 ("Probability distribution of monthly temperature and precipitation") and 3.2 ("TOA net radiation flux") to present the radiative forcing results firstly. In section 2 ("Data and Methods") we refer to Schmidt et al. (2012) and Kashimura et al. (2017) to show the large differences in forcing exerted by G1 solar dimming and G4 stratospheric aerosol geoengineering. In this paragraph we give the numbers of net radiative flux reduction due to G1 solar dimming and G4 stratospheric aerosol geoengineering methods. Please refer to our previous replies to your general comments.

The PDFs results in our study are different to Curry et al. (2014). Curry et al. (2014) use the piControl as the reference experiment and compare the PDFs of G1 with piControl, which suggests temperature and precipitation perturbations that occur under abrupt4 × CO2 are all reduced to near-piControl values by G1-type geoengineering. In our study, we choose abrupt4×CO2 as the reference for G1 and rcp45 as the reference for G4. We try to show how the global mean and extreme temperature and precipitation can be ameliorated by G1 solar dimming and G4 stratospheric aerosol injection. We add the following to clarify:

The PDFs for G1 and abrupt4×CO2 differ from those presented by Curry et al. (2014) as expected, due to the different choice of reference simulation. Curry et al. (2014) use the piControl as the reference experiment and compare the PDFs of G1 with piControl, which suggests temperature and precipitation perturbations that occur under abrupt4×CO2 are all reduced to near-piControl values by G1 solar dimming geoengineering. In our study, we choose abrupt4×CO2 as the reference for G1, and rcp45 as the reference for G4, as we aim to investigate how the global mean and extreme temperatures and precipitation events may be ameliorated by G1 solar dimming and G4 stratospheric aerosol injection geoengineering compared to global warming.

*Line 14: TOA forcing (SW +LW) is more general than SW at the surface. For sulfate CE the LW part is important. In case TOA forcing adds some information, you may add this to Figure 2.*

Reply: Thanks. We revise the Figure 2 to show the differences of TOA SW+LW forcing. At the same time, the panel (c) shows the difference between normalized TOA net radiation fluxes of G1-abrupt4xCO2 and G4-rcp45, instead of non-normalized ratios between G1-abrupt4xCO2 and G4-rcp45. As we reorder the sections 3.1 and 3.2, this new figure is labeled as Figure 1. We define the normalization methods in Section 2: Data and methods. Please refer to our previous replies on normalization methods.

[Figure]

**Figure 1:** Geographical distributions over the 40-year analysis periods of differences in net radiation flux at TOA between G1-abrupt4×CO2 (top), G4-rcp45 (middle). The bottom panel shows the differences in net radiation flux at TOA between normalized G1-abrupt4×CO2 and G4-rcp45 Stippling indicates regions where fewer than 5 of 6 models agree on the sign of the model response. The right sub-panels show the zonal average of the left sub-panels. Note that all three panels have different scales.

We also revise the relevant section (3.1 TOA net radiation) as the following:

[revised manuscript text omitted]

Pitari, G., Aquila, V., Kravitz, B., Robock, A., Watanabe, S., Cionni, I., Luca, N. D., Genova, G. D., Mancini, E., and Tilmes, S.: Stratospheric ozone response to sulfate geoengineering: Results from the Geoengineering Model Intercomparison Project (GeoMIP), J. Geophys. Res.-Atmos., 119, 2629–2653, 2014.

*Line 25: In case you talk about single models you may take into account that the forcing from the sulfate layer may be very different between the models (see (Pitari et al., 2013))*

Reply: Thanks. We add the following sentences in the revised section 3.1 on TOA net radiation:
The G1 solar dimming assumes global uniform solar reduction, while under G4 sulphate aerosols are handled differently among the participating models. GISS-E2-R and HadGEM2-ES adopt stratospheric aerosol schemes to simulate the sulfate aerosol optical depth (AOD), BNU-ESM and MIROC-ESM use the prescribed meridional distribution of AOD recommended by the GeoMIP protocol, CanESM2 specifies the uniform sulfate AOD (Kashimura et al., 2017). NorESM1-M specifies the AOD and effective radius which were calculated in previous simulations with the aerosol microphysical model ECHAM5-HAM (Niemeier et al., 2011). Although a prescribed AOD can be set, difference in assumed particle size for the stratospheric sulfate aerosols (Pierce et al., 2010) and the warming effects of stratospheric aerosol (Pitari et al., 2014) cause difference in the SRM forcing.

*Page 9*
*Line 2/3: Climate under G1 is pre-industrial while G4 is 2020. This can play a role here.*

Reply: Yes, we revise this sentence as following:
It is also possible that the different mean climate states between G1 and G4, and surface albedo changes due to sea ice and snow cover are responsible for the large differences in net radiation flux in the coastal Antarctic seas, and the more modest differences seen in the North Atlantic and Barents Sea along with Alaska and eastern Siberia.

*Figure 2: I wonder about the hatching in c). Only small areas with values close to one are significant. Is this correct and do you have an explanation?*

Reply: In the previous Figure 2c, we defined the significant change as larger than the 95th or smaller than the 5th percentile threshold value of the ratios between non-normalized G1-abrupt4xCO2 and G4-rcp45 for TOA net shortwave radiation fluxes from all model grids. In the new revised figure (Figure 1 in the revised manuscript), we show the TOA SW+LW forcing of G1-abrupt4xCO2 and G4-rcp45 in Figure 1a and 1b, and the differences between

normalized G1-abrupt4xCO2 and G4-rcp45 in Figure 1c. Now the stippling indicates regions where fewer than 5 of 6 models agree on the sign of the model response. By showing the differences between normalized G1-abrupt4xCO2 and G4-rcp45, we can better present the non-uniform regional responses between G1 solar dimming and G4 stratospheric aerosol geoengineering methods.

In the new Figure 1c, we find G4 stratospheric aerosol geoengineering introduces a more effective reduction in TOA net radiation over northern hemisphere, especially over the high-latitude continents, such as north of Northern America, Siberia and some regions of western Europe. This pattern of consistent responses in new Figure 1c is similar to hatching areas where the ratios are significantly small (therefore large reduction in net radiation in G4) in the previous Figure 2c.

*Page 11*
*Figure 3: You may add some words that Rx5days and G4 is partly positive, increase in precipitation, which indicates nicely the climate variability.*

Reply: Thanks. We add a sentence to clarify it:
In contrast, the index for G4-rcp45 (Fig.3f) is near-zero, though slightly negative on the whole, with the multi-model mean value of -1.8±0.9 mm. The partly positive Rx5day for G4-rcp45 reflects the climate variability simulated by models and lower signal-to-noise ratio.

*Page 12*
*Line 2: Do mean values of p and t scale the same in G1 and G4?*

Reply: The scaling of mean precipitation and mean temperature is not same in G1 and G4. But the scaling of extreme precipitation represented by Rx5Dday and mean temperature is almost same if one model is excluded. We elaborate these in the main text as following:
If relative humidity and atmospheric circulation remain relatively unchanged, then intense precipitation amount is governed by total precipitable water in the atmosphere, which the Clausius–Clapeyron relation says scales with mean temperatures (Allen and Ingram, 2002). The global mean precipitation decreases 2.1±0.4% per Kelvin in response to G1 solar dimming, and 2.7±1.0% per Kelvin in response to G4 stratospheric aerosol injection. The GISS-E2-R model contributes a relatively large portion to the spread of scaling between mean precipitation and temperature with a value of 4.5% per Kelvin for G4. If excluding the GISS-E2-R model, the global mean precipitation decreases 2.0±0.4% per Kelvin in response to G1 solar dimming, and 2.3±0.5% per Kelvin in response to G4 stratospheric aerosol injection. The scaling between mean precipitation and mean temperature under G1 and G4 is smaller than 3.4% precipitation change per Kelvin estimated from other coupled models under long-term equilibrium climate in response to doubling $CO_2$ (Allen and Ingram, 2002). The global mean Rx5day decreases 3.4±1.0% per Kelvin in response to G1 solar dimming, and 4.3±2.6% per Kelvin in response to G4 stratospheric aerosol injection. GISS-E2-R gives global mean Rx5day decreases 9.5% per Kelvin for G4. If excluding GISS-E2-R model, the global mean Rx5day decreases 3.4±1.1% per Kelvin in response to G1 solar dimming, and 3.3±0.6% per Kelvin in response to G4 stratospheric aerosol injection. The scaling of mean precipitation and mean temperature is expected to be much less than the 6.5% per Kelvin implied by the Clausius–Clapeyron relation, as the global-mean precipitation is primarily constrained by the availability of energy not moisture (Pall et al., 2007). The scaling of Rx5day and mean temperature under G1 and G4 is close to, but still

weaker than the Clausius–Clapeyron relation, probably because Rx5day is not really an index of the heaviest rainfall events that are expected to be constrained by the Clausius–Clapeyron relation. The Clausius–Clapeyron relation implies the same scaling of extreme precipitation and mean temperatures under both G1 and G4 experiments, which is the case here for five of six models, but not the GISS-E2-R model.

Allen, M. and Ingram, W.: Constraints on future changes in climate and the hydrologic cycle, Nature, 419, 224–232, 2002.

Pall, P., Allen, M. R., and Stone, D. A.: Testing the Clausius–Clapeyron constraint on changes in extreme precipitation under CO2 warming, Climate Dyn., 28, 351–363, doi:10.1007/ s00382-006-0180-2, 2007.

*Fig 4: Please add normalized values here. You may normalize with mean T and P or with the TOA forcing difference G1-4xCO2 and G4-RCP45.*
*I am not sure I understand what you did in Fig 4i) You may add an equation. Again, hatching and significance seem not to fit. Are this area not significant? Otherwise your area with significant results is very small and you should discuss the relevance of your results in this case.*

Reply: Thanks. Now we show normalized differences between G1-abrupt4xCO2 and G4-rcp45 in Figure 4c, 4f and 4i instead of ratios of non-normalized G1-abrupt4xCO2 and G4-rcp45. Stippling in all single figures indicate regions where fewer than 5 of 6 models agree on the sign of the model response. Although the old Figure 4c, 4f and 4i use the hatching indicates significant change (larger than the 95th or smaller than the 5th percentile threshold value), the regions of significant change are similar to the new Figure 4c, 4f and 4i where models show consistent responses. For example, Northern North America, Western Europe in Figure 4c, Central Europe and Siberia in Figure 4f.

In the main text, we add the following sentences to clarify:
The stratospheric aerosol injection more effectively reduces the TNn in northern North America and western Europe compared with solar dimming, while the solar dimming more effectively reduces TNn in the Siberian coastal region, Eastern Antarctica and the adjacent ocean regions. The stratospheric aerosol also effectively reduces the TXx in northern North America and central Europe compared with solar dimming, but with a smaller spatial extent and magnitude compared with TNn.  Stratospheric aerosol is more effective at reducing TXx in the Siberian coastal region, while the solar dimming seems more effective on reducing TNn there. ……
The difference between normalized change of G1-abrupt4×CO2 and G4-rcp45 is noisy and without coherent patterns (Fig. 4i).

[Figure]

**Figure 4: Geographical distributions over the 40-year analysis periods of the differences G1 - abrupt4×CO2 (left column), G4 - rcp45 (middle column), and differences between normalized G1 - abrupt4×CO2 and G4 - rcp45 (right column) for the extreme indices TNn (top row), TXx (middle row), and Rx5day (bottom row). Stippling indicates regions where fewer than 5 of 6 models agree on the sign of the model response. Note that panels have different colour scales.**

*Page 13*
*Line 6 to 16: You give a list of related topics here but I miss the clear relation to your results.*

Reply: The discussion in this section refers to reasons why we see "the signature of polar amplification evident in both hemispheres but primarily in the Arctic" (lines 4-5 on page 13). We try to resolve this by modifying the text to:

The cooling patterns seen for TNn (Fig. 4a,b) are similar but with a larger signal for G1-abrupt4×CO2 than G4-rcp45, with the signature of polar amplification evident in both hemispheres but primarily in the Arctic. Several studies have considered the reasons behind this effect.

*Line 17 to 24: The paper would gain of you can explain this feature better*

Reply: Thanks. We revise this paragraph to the following to better explain the land-sea contrast differences between TXx and TNn:

A notable feature in Fig. 4a, 4b, 4d and 4e is the larger cooling of TNn and TXx over land compared with oceans, also expressed in Table 3. The land-sea cooling contrast is larger for TXx than TNn (Fig 4d, e; Table 3), and TXx shows more uniform cooling than TNn across all latitudes. This feature is consistent with the stronger relationship of shortwave forcing to TXx. Under GHG warming scenarios, heat capacity differences, contrasts in surface sensible and latent fluxes, and boundary layer differences lead to contrasts opposite to those under G1 and G4

(Sutton et al., 2007; Joshi et al., 2008). Under G1 and G4, GHG warming occurs 24 hours a day, while reduced solar radiation is more effective in reducing day-time temperatures (TXx), with the land-sea heat capacity differences further enhancing TXx over TNx. The land–sea cooling effects under G4-rcp45 (Fig 4b,4e) are consistent with Volodin et al. (2011) who found increased land–sea cooling contrast in annual mean temperature using the INMCM model forced with 4 Mt S/year equatorial stratospheric aerosol injection.

*Line 28: 'Except Eastern China' this is a very small area. I guess this is a regional feature which is usually not very well represented in the models. Otherwise an explanation would be great.*

Reply: Agreed. We delete this part and revise the sentence to:
The pattern is similar in G4-rcp45 but with a smaller magnitude.

*Line 29: Better use solar dimming and stratospheric aerosol here also, so decide for one naming and stick to it.*

Reply: Agreed. We also change other places to make it consistent.

*Page 14*
*Line 11 to 13: I got lost. Can I see this in any figure?*

Reply: We revise this sentence to the following:
The ensemble means show that Rx5day is strongly reduced over equatorial regions, especially in the equatorial Pacific and southern flank of the Tibetan Plateau (Fig. 4g,h).
*Line 17/18: As you mention them, you may summarize the more complex interactions.*

Reply: Thanks. We revise the part as the following:
This has been attributed to a weaker Hadley cell due to weaker radiative forcing (Tilmes et al., 2009), but more recent analysis of the tropical circulation suggests more complex interactions between radiative forcing and Hadley cell extent and intensity. Under GeoMIP G1 experiment, the Hadley cell edges remain at their preindustrial width latitudinally, despite the residual stratospheric cooling associated with elevated carbon dioxide levels (Davis et al., 2016; Guo et al., 2018). The damping of the seasonal migration of the Intertropical Convergence Zone (ITCZ) within the Hadley cell under G1 is associated with preferential cooling of the summer hemisphere (Smyth et al., 2017).

*Line 28: Can you show this in a figure, link to existing figure or literature. In general, the small signal to noise ration is important for G4.*

Reply: Thanks. We elaborate this line and its context as the following:
Furthermore, monsoonal regions including East Asia and India exhibit a reduction in Rx5day under G1-abrupt4×CO2, which may be attributed to a weakened monsoon. Tilmes et al. (2013) observed, using a larger ensemble of models, that G1-abrupt4×CO2 results in a robust decrease in monsoonal precipitation, while it increases under abrupt4×CO2. Reduced Rx5day over monsoon regions is an indicator of weakened monsoon (Fig. 4g), because although the extreme precipitation index is calculated on an annual basis, it is dominated by wet

season precipitation, particularly in monsoon areas (Klein Tank et al., 2006). However, the change under G4-rcp45 is not as robust (Fig. 4h), due at least partially to lower mean temperature changes and land-sea thermal contrast, and therefore smaller signal-to-noise ratios compared with G1-abrupt4×CO2.

Klein Tank, A. M. G., Peterson, T. C., Quadir, D. A., Dorji, S., Zou, X., Tang, H., Santhosh, K., Joshi, U. R., Jaswal, A. K., Kolli, R. K., Sikder, A. B., Deshpande, N. R., Revadekar, J. V., Yeleuova, K., Vandasheva, S., Faleyeva, M., Gomboluudev, P., Budhathoki, K. P., Hussain, A., Afzaal, M., Chandrapala, L., Anvar, H., Amanmurad, D., Asanova, V. S., Jones, P. D., New, M. G., and Spektorman, T.: Changes in daily temperature and precipitation extremes in central and south Asia, J. Geophys. Res., 111, D16105, doi:10.1029/2005JD006316, 2006.

*Page 17*
*Line 1 to 5: The forcing varies stronger in G4 between the models than in G1. In case a model simulates of provides most sulfate in the tropics cooling at the poles would be relatively low. You may show TOA of the single models in an appendix.*

Reply: Yes. We show the TOA net radiation flux differences of G1-abrupt4×CO2 and G4-rcp45 for each model as Figure S1 and S2. We also add the following sentences to further clarity this point:

The spatial pattern of the TOA net radiation flux varies relatively more in G4-rcp45, ranging from -0.22 to -0.56 $Wm^{-2}$, while comparatively ranging from -2.29 to -3.40 $Wm^{-2}$ in G1-abrupt4×CO2 (Figure S1, S2). As simulation of sulphate aerosols differs among the participating G4 models, the spatially varying forcing results in very different cooling patterns particularly at high latitudes.

**TOA net radiation flux (G1-abrupt4xCO2)**

Figure S1: Geographical distributions over the 40-year analysis periods of the differences G1 - abrupt4×CO2 for TOA net radiation flux for BNU-ESM, CanESM2, GISS-E2-R, HadGEM2-ES, MIROC-ESM and NorESM1-M.

[Figure]

**Figure S2: Geographical distributions over the 40-year analysis periods of the differences G4 – rcp45 for TOA net radiation flux for BNU-ESM, CanESM2, GISS-E2-R, HadGEM2-ES, MIROC-ESM and NorESM1-M.**

*Line 9: Better cite Schmidt et al, 2012 or Kravitz et al, 2013 here. They have shown the shift of the ITCZ.*

Reply: Actually Kravitz et al. (2013) does not mention the effect, and Schmidt et al. (2012) notes that "While in HadGEM2-ES the zonally averaged position of the main branch of the ITCZ remains almost unchanged, it shifts slightly equatorward in the other three models" and "except for a small band slightly north of the equator over the Pacific which indicates an equatorward shift of the ITCZ in three of four models as mentioned above". They do not look at seasonality which is what we specifically talk about in this section, we change the text as:

This may be related to the reduced latitudinal extent of seasonal movement of the ITCZ under G1 as noted in previous studies (Schmidt et al., 2012; Smyth et al., 2017).

*Line 15: Any explanation for the different results over central Asia. This results is also different to Aswathy et al (2015).*

Reply: Aswathy et al. (2015) compared the sea-salt seeding experiments, sulfate injection experiments and RCP4.5 experiments during 2040-2069 to the RCP4.5 experiments during 2006-2035 control period, this is different from our comparison in this study, we compared the same period of 2030-2069 for G4 and RCP4.5. Aswathy et al. (2015) prescribe the radiative forcing to remain at the 2020 levels implied by the anthropogenic climate change under the RCP4.5. While in our study, we used the GeoMIP G4 simulation results, in which a constant 5Tg per year of $SO_2$ is introduced into the lower tropical stratosphere of climate models during the period of 2020-2069, while greenhouse gas forcing is defined by the RCP4.5 scenario.

In the previous version of this manuscript, we made a mistake in visually interpreting part of the results based on Figure 5e and 5f with Figure 4g and 4h. Now we reverse the colorbar of Figure 4g and 4h to make it clearer for comparing, and revise this paragraph as following:

The equatorial Pacific in the vicinity of the ITCZ displays increases in CDD under G1-abrupt4×CO2 at the same locations (Fig. 5e) as Rx5day decreases (Fig. 4g). This may be related to the reduced latitudinal extent of seasonal movement of the ITCZ under G1 as noted in previous studies (Schmidt et al., 2012; Smyth et al., 2017). Anti-correlation between CDD and Rx5day can also be seen for decreases in CDD and increases in Rx5day in the tropical Atlantic, South Atlantic and the southeast Pacific dry zone. Both northern and southern high latitudes, and large parts of Eurasia display increases in CDD and decreases in Rx5day (Fig. 5e, 4g). CDD decreases in the desert regions of northern Africa, southwestern Africa, Australia and southwestern North America, which are strongly influenced by the descending branch of the tropical Hadley cell. This implies most places have fewer droughts under the geoengineering simulation than without it. Fig. 5f shows that the pattern in G4-rcp45 is similar to G1-abrupt4×CO2 but noisier.

*Page 18*
*Fig 6: I am not sure I understood how you normalize the right column. I understood (i) as b/e and (j) as c-f. But than the pattern should be similar.*
*Increase the contrast of Fig (c) and (f), thus darker blue.*
*Can you add significance to (i) and (j)? Values around 1 and 0 should be white (also in Fig 7 and 8)*

Reply: Thanks. These suggestions largely improved our figures on showing monthly zonal differences between solar dimming and stratospheric aerosol geoengineering. Please check the new Figure 6, 7 and 8 in our replies to your general comments.

We remove the previous Fig (j) and plot the new Fig (i) as the difference between Fig (c) and Fig (f), Fig (c) and Fig (f) represent normalized response of solar dimming geoengineering relative to abrupt4xCO2 and stratospheric aerosol injection geoengineering relative to rcp45, so that (c) and (f) can be compared directly. We use darker blue or red for the right column sub-figures for Figure 6, 7 and 8. We also use white color to represent values around zero as you suggested.

*Page 19*
*Line 8: Which figure? Cannot see this in 6c) or 6f).*

Reply: We expand this sentence more clearly as the following:
The annual zonal mean pattern of G4-rcp45 (Fig. 6d) is comparable to G1-abrupt4×CO2 (Fig. 6a), but weaker by a factor of 7 to 9 in terms of their absolute magnitudes (Fig. 6g).

*Line 18: 'affected more by solar dimming....' Where? It is largely blue. I may not get all positive and negative changes right. You may explain this a bit more.*

Reply: In the newly revised Figure 6i, the differences in normalized G1-abrupt4×CO2 and G4-rcp45 (i.e. Fig. 6i) shows the difference between Fig. 6c and Fig. 6f. The negative values (blue colors) indicate relatively greater changes with G1, the positive values (red colors) indicate relatively greater changes with G4. Therefore, this sentence holds for the new Fig. 6i as well, and we further expand it to include spring season.

Fig. 6i shows that TNn in the northern high latitude springs and summers is affected much more by solar dimming than by stratospheric aerosol injection.

*Line 19: Quit dark red around 60S*

Reply: Thanks for pointing this out. We increase the contrast of Fig. 6i. In the new Fig. 6i, darker red represents relatively large change of G4 and darker blue represents large change of G1. Therefore, the blue occurring in the wintertime and springtime Southern Ocean around 60S suggests the TNn affected more by solar dimming than by aerosol injection, and it's similar to the springtime and summertime of northern high latitudes. We revise it as follows:

Fig. 6i shows that TNn in the northern high latitude springs and summers is affected much more by solar dimming than by stratospheric aerosol injection. A similar response is also present in the wintertime and springtime Southern Ocean. The only regions where stratospheric aerosol injection induces a significantly larger response than solar dimming is in the high Arctic in winter and latitudes between 40º-60ºN in spring and winter, suggestive of a longwave radiative effect of the aerosols.

*Page 21*

*Line 9 to 15: You are on quite weak ground here. The polar stratospheric vortex may weaken due to less sea ice and more wave propagation, sulfate warming the stratosphere may strengthen the vortex (Ferraro et al, 2014) etc.*

Reply: Yes. We revise this part as the following:

Fig. 7i suggests that the relative effectiveness of stratospheric aerosols and solar dimming is similar, except for the Arctic, and perhaps Antarctica, where aerosols appear more effective than dimming in winter. Since the lack of shortwave radiative forcing during winter would not lead to differences in solar dimming or aerosol response, atmospheric circulation changes are implicated. The tropical lower stratospheric radiative heating due to stratospheric aerosol would drive a thermal wind response, which would intensify the stratospheric polar vortices. In contrast, solar dimming does not produce this effect, and so there is little intensification of the polar vortex in G1. Therefore, the response of the northern hemisphere polar vortex to solar dimming geoengineering is much weaker than under stratospheric aerosol injection (Ferraro et al. 2015). A strengthening of the wintertime stratospheric polar vortices occurs under G4, tending to cool polar surface temperatures, which is consistent with wintertime northern hemisphere TNn and TXx patterns shown in Fig. 6i and 7i.

Ferraro, A. J., Charlton-Perez, A. J. and Highwood, E. J.: Stratospheric dynamics and midlatitude jets under geoengineering with space mirrors and sulfate and titania aerosols, J. Geophys. Res. Atmos., 120, 414–429, doi:10.1002/2014JD022734, 2015.

*Page 23*

*Line 16: Yes, but this depends also on the meridional distribution of the aerosols which is not given in the paper and may be different between the models.*

Reply: Agreed. We elaborate this:

Fig. 8g, 8h shows that the zonal means are noisier than for TNn and TXx. The results look much more complex than the temperature extreme indices in Fig. 6h and 7h. The general effect is that the tropical regions (30ºS-30ºN) are more strongly affected by aerosol injection than by solar dimming. The mid-latitude Rx5day is more effectively changed by stratospheric aerosol injection geoengineering year-round, especially in the Northern Hemisphere. Except for summertime polar areas, solar dimming geoengineering is relatively more effective year-round at high-latitudes, especially in the southern hemisphere. Ferraro et al. (2014) found that the tropical overturning circulation weakens in response to geoengineering with stratospheric sulphate aerosol injection due to radiative heating from the aerosol layer, but geoengineering simulated as a simple reduction in total solar irradiance do not capture this effect. Therefore, a relatively large tropical precipitation perturbation occurs under stratospheric aerosol injection. On the other hand, the meridional distribution of the sulfate aerosols is handled different between the models (as outlined in Section 3.1), which also contributes the noisier Rx5Day pattern showing in Fig. 8d, 8g and 8i. Four of the six models (BNU-ESM, CanESM2, MIROC-ESM and NorESM1-M) analysed in our study use the AOD prescribed to mimic the one-fourth of the 1991 eruption of Mount Pinatubo, but with different AOD meridional distribution, particle effective radii, and standard deviations of their log-normal size distribution (Kashimura et al., 2017). Another two models (GISS-E2-R and HadGEM2-ES) adopt different stratospheric aerosol schemes to simulate the sulfate AOD.

*Line 21: 'at high latitudes winter solar dimming ...' How can this be? There is no sunlight, so no SW reduction. < Line 23/24: 'Stratospheric vortex' I doubt that you see this with significant signals in the models even it seems to be a reasonable explanation. E.g. Driscoll et al. (2012) showed that climate model do not represent stratospheric dynamic repose of volcanic eruptions very well. You may show the zonal wind (DJF) and significance of the single models to approve the hypothesis.*

Reply: Thanks. We revise this part and deleted the sentence on speculating stratospheric vortex change as following:

The mid-latitude Rx5day is more effectively changed by stratospheric aerosol injection geoengineering year-round, especially in the Northern Hemisphere. Except for summertime polar areas, solar dimming geoengineering is relatively more effective year-round at high-latitudes, especially in the Southern Hemisphere.

*Page 24*
*Line 1 to 6: Does this result it to Jones et al. (2017)?*

Reply: No. Jones et al. (2017) found the stratospheric aerosol injection (SAI) applied to the southern hemisphere **only** would enhance tropical cyclone frequency relative to a global SAI application, and vice versa for SAI in the northern hemisphere **only**. Here our study of **global** geoengineering indicates the tropical storms and hurricanes are more effectively moderated in the Northern Hemisphere by global SAI and in the Southern Hemisphere by global solar dimming.

---

## Author Comment (AC2) · 22 Jun 2018

**Response to Review of "Extreme temperature and precipitation response to solar dimming and stratospheric aerosol geoengineering" by D. Ji et al.**

We first thank the referee for his/her insightful comments, which helped us clarify and greatly improve the paper. In the reply, the referee's comments are in *italics*, our response is in normal and changes to the text are shown in blue.

*Anonymous Referee #2*

*General Comments: In this manuscript, the authors analyzed the extreme values of climate indicators under 2 different solar radiation management scenarios G1 and G4. They took extreme index by ETCCDI and applied it on temperature and precipitation. The authors tried to find the differences and similarities on the global impact of two SRM experiment. And also tried to analysis the differences among the model.*

*This manuscript is novel and further complete the understanding of SRM. The structure is also well organized. I recommend the manuscript for publication though some of the comments still should be fixed or rephrased.*

*Specific Comments:*

*1. The significant regions in Fig. 2c and 4c,f,j need further descriptions on calculation process;*

Reply: Yes. We've revised the previous Figure 2 and Figure 4. In our new Figure 1 and Figure 4 (we reorder some paragraph, previous Fig.2 is labelled as Fig. 1 now, please refer to our replies to Referee #1), we show the normalized results. We normalize the values of each grid from the differences of G1-abrupt4xCO2 according to the global average of G1-abrupt4xCO2, same for G4-rcp45. With these normalized results, we present the difference between normalized G1-abrupt4xCO2 and G4-rcp45 instead of the ratio between non-normalized G1-abrupt4xCO2 and G4-rcp45 to avoid large unrealistic values. In Figure 6, 7 and 8, we also show the differences of zonally normalized results in several single figures instead of ratios between non-normalized fields. We define all normalization methods in Section 2: Data and methods as the following:

2.3 Normalization methods

There are large differences in forcing between the G1 solar dimming and G4 stratospheric aerosol injection geoengineering schemes. The mean and extreme climates under the two type geoengineering are quite different as will be shown below. To aid the comparisons, we adopt the following normalization methods to compare spatially relative effectivities between solar dimming and stratospheric aerosol injection.

The normalized global spatial effects of solar dimming or stratospheric aerosol injection are defined as the grid mean difference relative to the global mean difference:

$$< X^{geo} - X^{ref} > = \frac{\overline{X}_{grid}^{geo} - \overline{X}_{grid}^{ref}}{|\overline{X}_{global}^{geo} - \overline{X}_{global}^{ref}|}$$

where the operator $\diamond$ denotes the normalized grid value, X is TXx, TNn, Rx5day or other climate field, an overbar denotes the average of each grid cell or the global average, the absolute operator $||$ in the denominator of the right term preserves the sign of the geoengineering anomaly. The superscript "geo" represents geoengineering experiments of G1 solar dimming or G4 stratospheric aerosol injection, the superscript "ref" represents the reference experiments of abrupt4×CO2 or rcp45.

To normalize zonal mean difference in the climate extreme indices relative to the global mean difference, we use a similar formula:

$$< X^{geo} - X^{ref} > = \frac{\overline{X}^{geo}_{zonal} - \overline{X}^{ref}_{zonal}}{|\overline{X}^{geo}_{global} - \overline{X}^{ref}_{global}|}$$

where the operator $\diamond$ denotes the normalized zonal mean, an overbar denotes the zonal or global average, the absolute operator $||$ in the denominator of the right term preserves the sign of the geoengineering anomaly.

[Figure]

**Figure 1: Geographical distributions over the 40-year analysis periods of differences in net radiation flux at TOA between G1-abrupt4×CO2 (top), G4-rcp45 (middle). The bottom panel shows the differences in net radiation flux at TOA between normalized G1-abrupt4×CO2 and G4-rcp45 Stippling indicates regions where fewer than 5 of 6 models agree on the sign of the model response. The right sub-panels show the zonal average of the left sub-panels. Note that all three panels have different scales.**

[Figure]

**Figure 4: Geographical distributions over the 40-year analysis periods of the differences G1 - abrupt4×CO2 (left column), G4 - rcp45 (middle column), and differences between normalized G1 - abrupt4×CO2 and G4 - rcp45 (right column) for the extreme indices TNn (top row), TXx (middle row), and Rx5day (bottom row). Stippling indicates regions where fewer than 5 of 6 models agree on the sign of the model response. Note that panels have different colour scales.**

*2. The uncertainty reason present on abstract may not be proper here. Rephrase the word may be better.*

Reply: Thanks. We revise the abstract as the following:

We examine extreme temperature and precipitation under two potential geoengineering methods forming part of the Geoengineering Model Intercomparison Project (GeoMIP). The solar dimming experiment G1 is designed to completely offset the global mean radiative forcing due to a $CO_2$-quadrupling experiment (abrupt4×CO2), while in GeoMIP experiment G4, the radiative forcing due to the representative concentration pathway 4.5 (RCP4.5) scenario is partly offset by a simulated layer of aerosols in the stratosphere. Both G1 and G4 geoengineering simulations lead to lower minimum temperatures (TNn) at higher latitudes, and on land primarily through feedback effects involving high latitude processes such as snow cover, sea ice and soil moisture. There is larger cooling of TNn and maximum temperatures (TXx) over land compared with oceans, and the land-sea cooling contrast is larger for TXx than TNn. Maximum 5-day precipitation (Rx5day) increases over subtropical oceans, whereas warm spells decrease markedly in the tropics, and the number of consecutive dry days decreases in most deserts. The precipitation during the tropical cyclone (hurricane) seasons becomes less intense, whilst the remainder of the year becomes wetter. Stratospheric aerosol injection is more effective than solar dimming in moderating extreme precipitation (and flooding). Despite the magnitude of the radiative forcing applied in G1 being ~7.7 times larger than in G4, and differences in the aerosol chemistry and transport schemes amongst the models, the two types of geoengineering show similar spatial patterns in normalized differences of extreme

temperatures changes. Large differences mainly occur at northern high latitudes, where stratospheric aerosol injection more effectively reduces TNn and TXx. While the pattern of normalized differences of extreme precipitation is more complex than that of extreme temperatures, generally stratospheric aerosol injection is more effective in reducing tropical Rx5day, while solar dimming is more effective over extra-tropical regions.

*3. On Page 15, Line 1-6, this paragraph are not linked so well with the context. There are also no further analysis on the daily rain types. Further explanation and graphs would be better.*

Reply: Thanks. As this is at the end of "3.4 Spatial Response in Extremes", and the tropical precipitation change constitutes a large percentage of global precipitation change, therefore we would like to address how the tropical precipitation change in response to G1 solar dimming and G4 stratospheric aerosol injection in different major rain types. To make it clear, we add the following sentence:

As the tropical extreme precipitation change constitutes a large percentage of global extreme precipitation change in response to two type geoengineering schemes (Fig. 4g, 4h), it is interesting to know how the G1 solar dimming and G4 stratospheric aerosol injection affect major rain types in tropical regions.

*Minor comments:*

*1. P2 L18: Missed ref. Lathan et al. 2. P3 L12: Missed ref. Niemeier et al.*

Reply: Thanks. We add missing references, such as Latham (1990), Niemeier et al. (2013), Pitari et al. (2014), Smyth et al. (2017):

Latham, J.: Control of global warming?, Nature, 347, 339–340, 1990.

Niemeier, U., Schmidt, H., Alterskjær, K., and Kristjánsson, J. E.: Solar irradiance reduction via climate engineering: Impact of different techniques on the energy balance and the hydrological cycle, J. Geophys. Res. Atmos., 118, 11905–11917, doi:10.1002/2013JD020445, 2013.

Pitari, G., Aquila, V., Kravitz, B., Robock, A., Watanabe, S., Cionni, I., Luca, N. D., Genova, G. D., Mancini, E., and Tilmes, S.: Stratospheric ozone response to sulfate geoengineering: Results from the Geoengineering Model Intercomparison Project (GeoMIP), J. Geophys. Res.-Atmos., 119, 2629–2653, 2014.

Smyth, J. E., Russotto, R. D., and Storelvmo, T.: Thermodynamic and dynamic responses of the hydrological cycle to solar dimming, Atmos. Chem. Phys., 17, 6439-6453, doi:10.5194/acp-17-6439-2017, 2017.

*3. P5 L1: The estimate of CSDI and WSDI is applied on ensemble mean temperature or mean CSDI/WSDI?*

Reply: The CSDI and WSDI are calculated for each model firstly, then equal weight is given to each model before calculating multi-model ensemble mean. We clarify this point in "Data and Methods" as following:

Equal weight is given to each model in the analysis, and climate extreme indices are calculated for each model before multi-model ensemble averaging is done.

*4. P8 L24-27: It is not clear for me about the relations between different models and the geoengineering impact. Further expression would be better.*

Reply: Yes, Thanks for this comment. In the revised manuscript we emphasize the differences between G1 solar dimming and G4 stratospheric aerosol injection, and how each model implements the G4 experiment.

In the "Introduction" section, we add previous studies discussing the differences of the two type geoengineering schemes:

Both methods would cool Earth's surface by reducing sunlight reaching the surface, either by aerosols reflecting sunlight or by artificially reducing the solar constant in climate models. The injected stratospheric aerosols under G4 not only scatter shortwave radiation, also absorb near infrared and longer wavelengths (Lohmann and Feichter, 2005). The differences between stratospheric aerosol injection and solar dimming are influenced strongly by the absorption of longwave radiation by aerosols, this atmospheric heating imbalance could further stabilize the troposphere and lead to stronger precipitation reduction under stratospheric aerosol injection than under solar dimming (Niemeier et al., 2013). That there can be a difference in the mean climate response in reduced solar constant and increased stratospheric sulphate aerosols has been shown (Yu et al., 2015; Niemeier et al., 2013; Ferraro et al., 2014) and we expect that this will also be evident in the temperature and precipitation extremes.

In the "Results" section 3.1, we add the following to show the impacts of the two type geoengineering schemes on TOA net radiation flux:

The forcing of the G1 solar dimming and G4 stratospheric aerosol injection experiments are quite different, there can be a difference in the mean and extreme climate responses. The multi-model ensemble mean net radiation flux at the top of atmosphere (TOA) is 2.76 $Wm^{-2}$ and 0.004 $Wm^{-2}$ for the abrupt×4CO2 and G1 experiments, and 1.63 $Wm^{-2}$ and 1.27 $Wm^{-2}$ for the rcp45 and G4 experiments during their 40-year analysis periods. Therefore, the G1 solar dimming and G4 stratospheric aerosol injection exert a reduction of 2.76 $Wm^{-2}$ and 0.36 $Wm^{-2}$ for net radiation fluxes at TOA respectively. The differences of mean net radiation flux at TOA over land and ocean between two geoengineering experiments and their reference experiments are show in Table 3. Although the ratio between the global temporally averaged net radiation flux reductions at TOA is a factor of ~7.7, the spatial distribution of net radiation flux changes for the G1 and G4 ensemble means are quite similar, especially the positive TOA net radiation over Greenland, Antarctica, North Africa and West Asia, and the negative TOA net radiation over North America, Central Europe and tropical ocean basins (Figure 1). The entire ensemble shows a large and consistent positive TOA net radiation east of Greenland in the North Atlantic under G1 solar dimming (Figure 1a), the region associated with the overturning part of the Atlantic meridional circulation (AMOC), and which under the G1 forcing was shown to be strongly affected by changes in radiative forcing and air/ocean heat exchange (Hong et al., 2017). However, differences are clearer when we investigate the spatial pattern of normalized effects exerted by the two SRM experiments, although most regions have differences close to zero for normalized solar dimming and stratospheric aerosol geoengineering effects on TOA net radiation (Figure 1c). The G4 stratospheric aerosol injection geoengineering introduces a more effective reduction in TOA net radiation over the Northern Hemisphere, especially over the high-latitude continents, such as northern North America, Siberia and some regions of western Europe. The G1 solar dimming geoengineering introduces a more effective reduction

in TOA net radiation over North Africa, northern South America, the Indian Ocean and tropical Western Pacific. In contrast, many other equatorial regions, the Southern Ocean and the Intertropical and South Pacific Convergence Zones display small differences between normalized solar dimming and stratospheric aerosol injection effects.

The G1 solar dimming assumes global uniform solar reduction, while under G4 sulphate aerosols are handled differently among the participating models. GISS-E2-R and HadGEM2-ES adopt stratospheric aerosol schemes to simulate the sulfate aerosol optical depth (AOD), BNU-ESM and MIROC-ESM use the prescribed meridional distribution of AOD recommended by the GeoMIP protocol, CanESM2 specifies uniform sulfate AOD (Kashimura et al., 2017). NorESM1-M specifies the AOD and effective radius which were calculated in previous simulations with the aerosol microphysical model ECHAM5-HAM (Niemeier et al., 2011, Niemeier and Timmreck, 2015). Although a prescribed AOD can be set, difference in assumed particle size for the stratospheric sulfate aerosols (Pierce et al., 2010) and the warming effects of stratospheric aerosol (Pitari et al., 2014) cause difference in the SRM forcing.

In "Results" section 3.2, we add following to show impacts of two type geoengineering schemes on mean climate states:

The G1 solar dimming and G4 stratospheric aerosol injection geoengineering greatly affected the mean climate states. The annual mean surface air temperatures are 291.0 K and 286.7 K for abrupt4×CO2 and G1 experiments, 288.8 K and 288.3 K for rcp45 and G4 experiments respectively during their 40-year analysis periods. The global hydrological strength is likewise reduced; the annual mean precipitation totals are 1125.8 mm and 1026.9 mm for abrupt4×CO2 and G1 experiments, 1098.4 mm and 1084.3 mm for rcp45 and G4 experiments (Table 3).

*5. P12 L2: May be I got missed but I'm not sure what the 'case' indicate.*

Reply: In our previous manuscript, the 'case' means the extreme precipitation scales with mean temperature. In the revised manuscript, we largely revise this paragraph as following:

If relative humidity and atmospheric circulation remain relatively unchanged, then intense precipitation amount is governed by total precipitable water in the atmosphere, which the Clausius–Clapeyron relation says scales with mean temperatures (Allen and Ingram, 2002). The global mean precipitation decreases 2.1±0.4% per Kelvin in response to G1 solar dimming, and 2.7±1.0% per Kelvin in response to G4 stratospheric aerosol injection. The GISS-E2-R model contributes a relatively large portion to the spread of scaling between mean precipitation and temperature with a value of 4.5% per Kelvin for G4. If excluding the GISS-E2-R model, the global mean precipitation decreases 2.0±0.4% per Kelvin in response to G1 solar dimming, and 2.3±0.5% per Kelvin in response to G4 stratospheric aerosol injection. The scaling between mean precipitation and mean temperature under G1 and G4 is smaller than 3.4% precipitation change per Kelvin estimated from other coupled models under long-term equilibrium climate in response to doubling $CO_2$ (Allen and Ingram, 2002). The global mean Rx5day decreases 3.4±1.0% per Kelvin in response to G1 solar dimming, and 4.3±2.6% per Kelvin in response to G4 stratospheric aerosol injection. GISS-E2-R gives global mean Rx5day decreases 9.5% per Kelvin for G4. If

excluding GISS-E2-R model, the global mean Rx5day decreases 3.4±1.1% per Kelvin in response to G1 solar dimming, and 3.3±0.6% per Kelvin in response to G4 stratospheric aerosol injection. The scaling of mean precipitation and mean temperature is expected to be much less than the 6.5% per Kelvin implied by the Clausius–Clapeyron relation, as the global-mean precipitation is primarily constrained by the availability of energy not moisture (Pall et al., 2007). The scaling of Rx5day and mean temperature under G1 and G4 is close to, but still weaker than the Clausius–Clapeyron relation, probably because Rx5day is not really an index of the heaviest rainfall events that are expected to be constrained by the Clausius–Clapeyron relation. The Clausius–Clapeyron relation implies the same scaling of extreme precipitation and mean temperatures under both G1 and G4 experiments, which is the case here for five of six models, but not the GISS-E2-R model.

*6. P13 L28: Eastern China in Fig4 seems no special around the globe, this part may need further explanation.*

Reply: We deleted this sentence. It's more likely a regional feature which is usually not very well represented in the models as suggested by Referee #1.

*7. P14 L10-13: The reduction of Rx5day is whether a result from Curry et al., 2014 or from the paper result? Further explanation would be better.*

Reply: Here we mean the results from the previous study by Curry et al. (2014). In our study, we also find the reduction of Rx5day under solar dimming and stratospheric aerosol injection geoengineering schemes. Please refer to the revised sentences following this line:

The ensemble means show that Rx5day is strongly reduced over equatorial regions, especially in the equatorial Pacific and southern flank of the Tibetan Plateau (Fig. 4g, 4h). This is due to increased atmospheric stability and suppression of convection under geoengineering (Bala et al.,2008).

*8. P15 L1-6: The paragraph may not fully link with the context and there is no graphs or tables to support the statistics.*

Reply: The numbers given are from simple calculations of the model precipitation output. We could have put them in a table but it seemed more concise to simply give the statistics as a sentence. The context comes because we are discussing Rx5day throughout the paragraph, and in particular tropical and monsoon rains (that is heavy rain).